# Prospects of a thousand-ion $Sn^{2+}$ Coulomb-crystal clock with sub-$10^{-19}$ inaccuracy

**David R. Leibrandt[1,2,3] ✉, Sergey G. Porsev ⬤[4], Charles Cheung ⬤[4] & Marianna S. Safronova ⬤[4]**

Optical atomic clocks are the most accurate and precise measurement devices of any kind, enabling advances in international timekeeping, Earth science, fundamental physics, and more. However, there is a fundamental tradeoff between accuracy and precision, where higher precision is achieved by using more atoms, but this comes at the cost of larger interactions between the atoms that limit the accuracy. Here, we propose a many-ion optical atomic clock based on three-dimensional Coulomb crystals of order one thousand $Sn^{2+}$ ions confined in a linear RF Paul trap with the potential to overcome this limitation. $Sn^{2+}$ has a unique combination of features that is not available in previously considered ions: a $^1S_0 \leftrightarrow {}^3P_0$ clock transition between two states with zero electronic and nuclear angular momentum ($I = J = F = 0$) making it immune to nonscalar perturbations, a negative differential polarizability making it possible to operate the trap in a manner such that the two dominant shifts for three-dimensional ion crystals cancel each other, and a laser-accessible transition suitable for direct laser cooling and state readout. We present calculations of the differential polarizability, other relevant atomic properties, and the motion of ions in large Coulomb crystals, in order to estimate the achievable accuracy and precision of $Sn^{2+}$ Coulomb-crystal clocks.

Optical atomic clocks based on single trapped ions and on ensembles of thousands of optical-lattice-trapped neutral atoms have both achieved $10^{-18}$ fractional frequency accuracy and precision[1], surpassing the microwave atomic clocks that underpin international atomic time (TAI) by three orders of magnitude, and enabling new technological applications such as relativistic geodesy[2] as well as tests of the fundamental laws of physics[3]. For any particular application, trapped-ion and optical-lattice clocks based on optical atomic transitions have complimentary advantages and limitations. Optical lattice clocks use simultaneous measurements of thousands of atoms to quickly average down quantum projection noise (QPN), a fundamental limit to clock precision, speeding up both characterizations of systematic effects and clock measurements, and increasing the bandwidth of sensing applications. However, they suffer from larger systematic frequency shifts due to blackbody radiation and interactions between atoms that

are difficult to control. Clocks based on one or a few trapped ions suffer from larger QPN, but offer exquisite control of environmental perturbations and interactions, high-fidelity universal quantum control at the individual ion level, and a wide variety of atomic, molecular, and highly charged ions to choose from with different sensitivities to environmental perturbations and proposed extensions of the Standard Model of particle physics.

Recently, a wide variety of new optical clock platforms, which overcome some of the limitations of conventional lattice and ion clocks, have been proposed and demonstrated. Composite optical clocks combining an ensemble of lattice-trapped atoms that provides high stability, together with a single trapped ion that provides high accuracy have been proposed[4,5], and the fundamental building blocks have been demonstrated[6]. Ion trap arrays have been built and shown to be capable of supporting clock operation with around 100 ions and

[1]Department of Physics and Astronomy, University of California, Los Angeles, CA 90095, USA. [2]Time and Frequency Division, National Institute of Standards and Technology, Boulder, CO 80305, USA. [3]Department of Physics, University of Colorado, Boulder, CO 80309, USA. [4]Department of Physics and Astronomy, University of Delaware, Newark, DE 19716, USA. ✉e-mail: leibrandt@ucla.edu

sub-$10^{-18}$ accuracy[7]. Clocks based on arrays of optical-tweezer-trapped atoms offer the prospect of individual-atom-resolved quantum control, and have demonstrated coherence times exceeding 40 s and operation with up to 150 atoms[8]. Three-dimensional optical lattice clocks greatly suppress the interactions between atoms present in current lattice clocks based on one and two-dimensional optical lattice traps and have demonstrated operation with up to $10^5$ atoms[9]. Finally, recent demonstrations have shown that it is possible to build clocks based on highly charged ions which can have lower sensitivity to environmental perturbations and higher sensitivity to physics beyond the Standard Model[10–13].

Three-dimensional Coulomb crystals of thousands of ions in linear RF Paul traps have been studied for many years for other applications[14–19]; however they were thought to be impractical for high accuracy atomic clocks because ions located off of the trap axis, along which the RF trapping electric field is zero, suffer from driven motion called micromotion that leads to large time-dilation systematic frequency shifts. Berkeland et al.[20] pointed out that for clock transitions with a negative differential polarizability, the negative micromotion time-dilation shift could be almost perfectly canceled out by a positive differential Stark-shift caused by the same RF trapping field that drives the micromotion, provided that the trap is driven at a "magic" value of the RF frequency. Building on this, Arnold et al.[21] proposed that many-ion clocks could be built based on three-dimensional Coulomb crystals of ions with negative-differential-polarizability clock transitions, which at the time was known to include $B^+$, $Ca^+$, $Sr^+$, $Ba^+$, $Ra^+$, $Er^{2+}$, $Tm^{3+}$, and $Lu^+$. $Lu^{2+}$[22] and $Pb^{2+}$[23] were later added to this list, and Kazakov et al.[24] performed a detailed analysis of micromotion shifts in these systems and proposed active optical clocks based on negative-differential-polarizability transitions. In contrast with earlier theoretical predictions, it was later experimentally determined that the $^1S_0 \leftrightarrow {}^3D_1$ clock transition of $Lu^+$ actually has a very small but positive differential polarizability. The alternative $^1S_0 \leftrightarrow {}^3D_2$ clock transition of $Lu^+$ was determined to have a negative but larger differential polarizability[25]. Of the remaining candidates for Coulomb-crystal clocks, $Ca^+$, $Sr^+$, $Ba^+$, $Ra^+$, $Er^{2+}$, $Tm^{3+}$, $Lu^+$ $^1S_0 \leftrightarrow {}^3D_2$, and $Lu^{2+}$ have comparatively short excited-state lifetimes and suffer from inhomogenous broadening due to quadrupole and tensor polarizability shifts[22]. Beloy et al.[23] proposed doubly ionized group-14 elements with $I = J = F = 0$ such as $Pb^{2+}$ as ideal trapped-ion clock candidates because their $^1S_0 \leftrightarrow {}^3P_0$ transitions are completely immune to these nonscalar perturbations.

$B^+$ and $Pb^{2+}$ have doubly-forbidden intercombination clock transitions with $> 10^3$ s lifetimes, but do not have transitions suitable for direct state readout. In $B^+$ the $\Gamma = 2\pi \times 2$ Hz decay rate of the $^1S_0 \leftrightarrow {}^3P_1$ transition is prohibitively slow, and in $Pb^{2+}$ the wavelength of this transition is 155 nm. The latter is below the low-wavelength cutoffs for generation of UV laser light by nonlinear frequency conversion in

workhorse materials $\beta$-$BaB_2O_4$ (BBO, 182 nm phase matching cutoff[26]), $CsLiB_6O_{10}$ (CLBO, 180 nm transparency cutoff[27]), and $LiB_3O_5$ (LBO, 160 nm transparency cutoff[28]). New candidate materials such as $KBe_2BO_3F_2$ (KBBF, 150 nm transparency cutoff[29]) and $BaMgF_4$ (BMF, 125 nm transparency cutoff[30]) are difficult to grow and therefore unavailable commercially with sufficiently high quality. Scaling indirect quantum-logic readout[31] to large Coulomb crystals remains an open challenge[32].

In this article, we propose Coulomb-crystal clocks based on of the order of one thousand $Sn^{2+}$ ions (see Fig. 1). We present a theoretical calculation of the differential polarizability of the $^1S_0 \leftrightarrow {}^3P_0$ clock transition, showing it to be both small and negative, meaning that the systematic shift due to blackbody radiation is small, and there is a magic trap frequency of 225(5) MHz at which the trap Stark shift cancels the micromotion time-dilation shift. Although this is a high trap frequency, previous experiments have operated in this range[33]. Similar to $Pb^{2+}$, in addition to having zero electronic angular momentum in the two clock states, there are several isotopes of $Sn^{2+}$ that have zero nuclear spin, and thus the clock transition is completely immune to nonscalar perturbations[23]. Furthermore, for these nuclear-spin-zero isotopes, the clock transition is an extremely forbidden intercombination transition with a zero-magnetic-field excited state lifetime on the order of years, but the lifetime can be tuned to an experimentally convenient value by applying a magnetic field[34,35]. However, unlike $Pb^{2+}$, $Sn^{2+}$ has a laser-accessible 181 nm $^1S_0 \leftrightarrow {}^3P_1$ transition suitable for direct laser cooling and state readout, enabling many-ion clock operation. We present calculations of the $Sn^{2+}$ atomic properties relevant for the evaluation of the systematic frequency shifts as well as calculations of the motion of ions in large Coulomb crystals that lead to imperfect cancellation of the micromotion time-dilation and trap Stark shift, and imperfections in the spectroscopic lineshape. It may be possible to simultaneously achieve $9.0 \times 10^{-20}$ total fractional inaccuracy and $4.4 \times 10^{-18}/\tau^{1/2}$ fractional imprecision, where $\tau$ is the measurement duration in seconds. We conclude by discussing prospects for fifth force searches using isotope shift measurements, as well as other tests of fundamental physics with $Sn^{2+}$.

## Results
### $Sn^{2+}$ atomic properties
Figure 2a shows an energy level diagram of $Sn^{2+}$. We carried out calculations of $Sn^{2+}$ atomic properties using a high-accuracy relativistic method combining configuration interaction (CI) with the linearized coupled-cluster approach (CI+all-order method)[36]. To evaluate uncertainties of the results, we also carried out the same calculations using a combination of CI and many-body perturbation theory (CI+MBPT method)[37]. Since CI is essentially complete for a divalent system, the main source of the uncertainty is the treatment of the core, which is

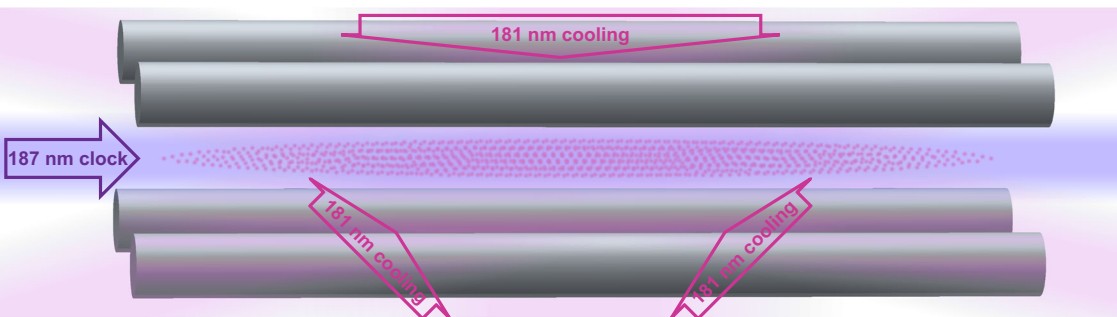

**Fig. 1 | Schematic of a many-ion optical atomic clock based on $Sn^{2+}$ ions.** A three-dimensional Coulomb crystal of 1000 $Sn^{2+}$ ions (small dots) is confined in a linear RF Paul trap (gray cylindrical electrodes). For single-ion secular motional frequencies 1.15 MHz, 0.85 MHz, and 0.10 MHz, the diameters of the ellipsoidal ion crystal are 19 $\mu m$, 44 $\mu m$, and 909 $\mu m$. $Sn^{2+}$ is laser-cooled using three orthogonal lasers (shown in pink) red-detuned from the 181 nm $^1S_0 \leftrightarrow {}^3P_1$ transition. Spectroscopy of the 187 nm $^1S_0 \leftrightarrow {}^3P_0$ clock transition is performed using a laser aligned with the axial trap direction (shown in purple), and readout is based on state-dependant photon scattering on the $^1S_0 \leftrightarrow {}^3P_1$ transition.

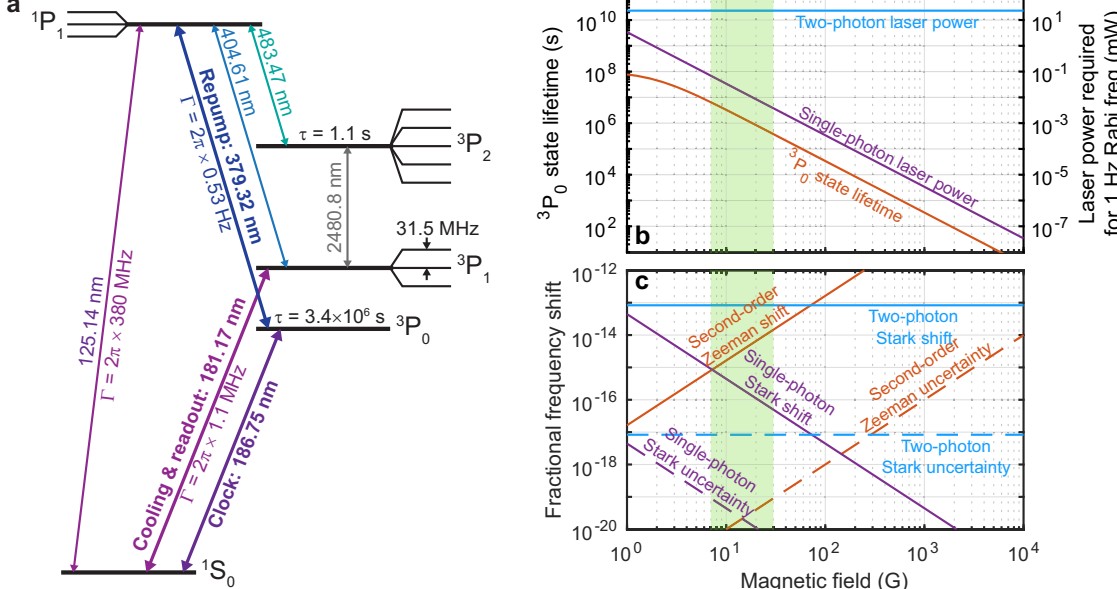

**Fig. 2 | Sn²⁺ atomic properties. a** Energy level diagram showing the states and transitions relevant for clock operation using a nuclear-spin-zero isotope of Sn²⁺ at a magnetic field of 15 G. Thick horizontal black lines indicate eigenstates of the magnitude of the orbital, spin, and total angular momentum labeled by the term symbols $^{2S+1}L_J$, where the quantum number $S \in \{0, 1\}$ is the magnitude of the spin angular momentum, $L \in \{S, P\}$ is the magnitude of the orbital angular momentum (where S denotes 0 and P denotes 1), and $J \in \{0, 1, 2\}$ is the magnitude of the total angular momentum. Thin black lines indicate Zeeman splittings of the states with different values of the $m$ quantum number, which denotes the projection of the total angular momentum on the quantization axis. **b** Lifetime of the excited clock state and the laser power required to drive the clock transition as a function of the applied magnetic field. At zero magnetic field, the clock transition is highly forbidden, leading to a spontaneous emission lifetime of the excited clock state of many years. As the magnetic field is increased, the $^3P_0$ state is weakly mixed with the $^{1,3}P_1$ states, leading to a reduced $^3P_0$ lifetime and laser intensity required to drive the transition. The laser power required to drive the clock transition with a Rabi frequency $\Omega/(2\pi) = 1$ Hz is shown for driving the transition with a single 187 nm photon, and for driving it with two 374 nm photons, assuming a 100 μm $1/e^2$ laser beam diameter. **c** Systematic frequency shifts and estimated uncertainties due to the second-order Zeeman effect caused by the applied magnetic field and the Stark effect caused by the laser used to drive the clock transition, as a function of the applied magnetic field.

estimated as the difference between the results obtained by these two methods[38]. The results of the calculations given in Tables 1 and 2 are used to design and evaluate the performance of the Sn²⁺ clock schemes.

The $5s5p\,^3P_1$ state mostly decays through the $^3P_1 \leftrightarrow {}^1S_0$ transition, which will be used for cooling and readout. We found this transition rate to be $7.04 \times 10^6\,\mathrm{s}^{-1}$ and, respectively, the lifetime of the $^3P_1$ state to be $\tau_{^3P_1} \approx 142$ ns. This lifetime scales like the transition wavelength cubed and is significantly shorter than other divalent atoms used for clocks primarily because of the shorter wavelength.

**Table 1 | Reduced matrix elements of the electric multipole $Ek$ operator (in $ea_0^k$), magnetic multipole $Mk$ operator (in $\mu_0\,a_0^{k-1}$), and the transition rates, $W$ (in s⁻¹), from 5s5p $^3P_{1,2}$ and 5s5p $^1P_1$ to the lower-lying states, calculated in the CI+all-order+RPA approximation**

| Transition | Mult. | ME | W (s⁻¹) |
|---|---|---|---|
| $^3P_1 - {}^1S_0$ | E1 | 0.249 | $7.04 \times 10^6$ |
| $^3P_1 - {}^3P_0$ | M1 | 1.41 | 0.080 |
| $^3P_2 - {}^1S_0$ | M2 | 11.2 | 0.027 |
| $^3P_2 - {}^3P_0$ | E2 | 5.06 | 0.0034 |
| $^3P_2 - {}^3P_1$ | M1 | 1.57 | 0.87 |
|  | E2 | 7.61 | 0.0014 |
| $^1P_1 - {}^1S_0$ | E1 | 2.64 | $2.40 \times 10^9$ |
| $^1P_1 - {}^3P_0$ | M1 | 0.141 | 3.3 |
| $^1P_1 - {}^3P_1$ | M1 | 0.122 | 2.0 |
|  | E2 | 1.16 | 0.47 |
| $^1P_1 - {}^3P_2$ | M1 | 0.159 | 2.0 |
|  | E2 | 0.873 | 0.11 |

The $^3P_0 \leftrightarrow {}^1S_0$ clock transition is forbidden by single-photon-transition selection rules, but it can be opened in the presence of the magnetic field due to the admixture of the $^{3,1}P_1$ states to $^3P_0$ by the $\boldsymbol{\mu} \cdot \mathbf{B}$ operator (where $\boldsymbol{\mu}$ is the magnetic dipole moment operator and $\mathbf{B}$ is the static magnetic field).

Restricting ourselves to the admixture of the two nearest to $^3P_0$ states of the same parity, $5s5p\,^3P_1$ and $5s5p\,^1P_1$, we arrive at the following expression for the $^3P_0 \leftrightarrow {}^1S_0$ transition rate:

$$W \approx \frac{2\omega^3}{27\varepsilon_0 hc^3} \times \left| \sum_{n = {}^3P_1, {}^1P_1} \frac{\langle {}^1S_0 ||d|| n \rangle \langle n ||\mu|| {}^3P_0 \rangle}{E(n) - E({}^3P_0)} \right|^2 B^2, \qquad (1)$$

where $\omega$ is the $^3P_0 \leftrightarrow {}^1S_0$ transition frequency, $\mathbf{d}$ is the electric dipole operator, $E$ is a state energy, and $\varepsilon_0, h = 2\pi\hbar$, and $c$ are the vacuum permittivity, Planck's constant, and speed of light, respectively. The matrix elements of the electric and magnetic dipole operators expressed in $ea_0$ and $\mu_0$ (where $e$ is the elementary charge and $a_0$ and $\mu_0$ are the Bohr radius and magneton) are given in Table 1. Using Eq. (1), we find the $^3P_0$ lifetime to be $\tau_{^3P_0} \approx 3.4 \times 10^8\,(\mathrm{s\,G}^2)/B^2$, where $1\,\mathrm{G} = 10^{-4}$ T. The CI+all-order values of the matrix elements and experimental energies[39] are used to compute all transition rates.

Static and dynamic polarizabilities at the clock wavelengths $\lambda_0 \approx 186.75$ nm and $\lambda = 2\lambda_0 \approx 373.5$ nm, needed for the evaluation of the blackbody radiation shift and the ac Stark shifts, are listed in Table 2. The blackbody radiation shift is mostly determined by the differential static polarizability of the $5s5p\,^3P_0$ and $5s^2\,{}^1S_0$ clock states, $\Delta\alpha \equiv \alpha(^3P_0) - \alpha(^1S_0) \approx -0.96(4)\,a_0^3$. Based on the difference between the CI+all-order and CI+MBPT values, we determined the uncertainty of $\Delta\alpha$. For the static polarizabilities it was found at the level of 4%.

**Table 2 | Static and dynamic polarizibilities of the $^3P_0$ and $^1S_0$ states and their differential polarizabilities Δα (in $a_0^3$), calculated in the CI+MBPT and CI+all-order (labeled as "CI+All") approximations**

| | | CI+MBPT | CI+All |
|---|---|---|---|
| Static | $\alpha(^1S_0)$ | 15.16 | 15.20 |
| | $\alpha(^3P_0)$ | 14.20 | 14.25 |
| | $\Delta\alpha$ | − 0.96(4) | − 0.96(4) |
| λ = 186.75 nm | $\alpha(^1S_0)$ | 27.18 | 27.80 |
| | $\alpha(^3P_0)$ | 23.30 | 23.42 |
| | $\Delta\alpha$ | − 3.9 | − 4.4(5) |
| λ = 373.5 nm | $\alpha(^1S_0)$ | 17.17 | 17.21 |
| | $\alpha(^3P_0)$ | 15.94 | 16.00 |
| | $\Delta\alpha$ | − 1.23 | − 1.21(2) |

Uncertainties are given in parentheses.

Micromotion driven by the rf-trapping field leads to ac Stark and second-order Doppler shifts. As predicted by Berkeland et al.[20] and demonstrated by Dubé et al.[40], if the differential static polarizability Δα for the clock transition is negative, there is a "magic" trap drive frequency $\Omega_m$ given by

$$\Omega_m = \frac{Q_i}{M_i c} \sqrt{-\frac{\hbar\omega}{\Delta\alpha}} \qquad (2)$$

($Q_i$ and $M_i$ are the ion charge and mass), at which the micromotion shift vanishes. All even isotopes of $Sn^{2+}$ are equally suitable for clock operation, and different choices will have slightly different magic drive frequencies and laser cooling dynamics. The calculations throughout this article use $M_i \approx A m_p$ with $A = 118$ (where $m_p$ is the proton mass), resulting in $\Omega_m/(2\pi) = 225(5)$ MHz.

Calculations of other atomic properties relevant for clock operation are described in the Methods.

**$Sn^{2+}$ clock operation**

Each cycle of basic $Sn^{2+}$ clock operation consists of four steps. First, the ions are laser cooled on the 181 nm $^1S_0 \leftrightarrow {}^3P_1$ transition. The $\Gamma/(2\pi) = 1.1$ MHz natural linewidth of this transition is sufficiently narrow for Doppler cooling to sub-mK temperatures, at which the fractional time-dilation shift due to secular (i.e., thermal) motion of the ions is at or below the $10^{-18}$ level, and can be characterized such that the uncertainty is significantly better. It is also likely to be sufficiently broad to achieve the high cooling rates necessary for the crystallization of ions initially loaded into the trap at above room temperature in a delocalized cloud phase[41]. For many-ion crystals with a spectrum of secular-motion normal mode frequencies spanning from tens of kHz to above 1 MHz, efficient Doppler cooling of all modes will require driving above saturation to power broaden the transition to significantly more than the motional mode bandwidth, and the use of three laser beams with k-vectors that have significant overlap with all spatial directions. In the calculations below, we assume a saturation parameter of 20, such that the transition is power broadened to a linewidth of 5 MHz.

Second, spectroscopy is performed on the $^1S_0 \leftrightarrow {}^3P_0$ clock transition. This transition can be driven either as a single-photon transition using a laser at 187 nm or as a two-photon transition using a laser at 374 nm (see Fig. 2b and c). The two-photon option has the advantages that it can be driven at or near zero applied magnetic field, which minimizes the second-order Zeeman shift, and using a photon from each of two counter-propagating beams, which eliminates motional sideband transitions, but the disadvantage that it requires high laser intensity with an associated Stark shift of order $10^{-13}$, and it may not be

possible to reduce the uncertainty to below $10^{-18}$. The single-photon option has the advantage that the required laser intensity can be reduced by increasing the magnetic field, but this comes at the cost of an increased second-order Zeeman shift. We estimate that at magnetic fields around 15 G, the quadrature sum of the uncertainties due to the probe-laser Stark shift and the second-order Zeeman shift can be minimized as shown in Fig. 2c. With 15 $\mu$W of laser power focused to a 100 $\mu$m $1/e^2$ beam diameter polarized parallel to the magnetic field direction, the single-photon Rabi frequency $\Omega/(2\pi) = 1$ Hz and the Stark shift is $2.0 \times 10^{-16}$. We conservatively estimate that this can be characterized with an uncertainty of $2.0 \times 10^{-20}$ by using hyper-Ramsey[42] or auto-balanced Ramsey[43] interrogation protocols. The second-order Zeeman shift is $-3.7 \times 10^{-15}$, and by measuring the clock transition frequency at a significantly higher magnetic field, it should be possible to characterize this shift to an uncertainty of $2.3 \times 10^{-20}$. For 1000 $Sn^{2+}$ ions in a trap with single-ion secular frequencies 1.15 MHz, 0.85 MHz, and 0.10 MHz, the diameters of the ellipsoidal ion crystal major axes are 19 $\mu$m, 44 $\mu$m, and 909 $\mu$m. Magnetic field inhomogeneity of 1 mG over this volume is sufficient to suppress inhomogeneous broadening due to the second-order Zeeman shift to a negligible level. Although it would be difficult to apply a 15 G quantization field with such a $10^{-4}$/mm fractional magnetic field gradient using standard Helmholtz coils, gradients as small as $10^{-6}$/mm are routinely achieved using solenoids together with shim coils[44]. The ion trap should be designed to minimize the capacitance between electrodes in order to to minimize the contribution to the second-order Zeeman shift due to the ac magnetic field at the trap drive frequency[45] (see Methods).

Third, the number of ions in the ground and excited states is read out by driving the $^1S_0 \leftrightarrow {}^3P_1$ transition and counting the total number of photons scattered into a photomultiplier tube (PMT) or other detector. During a $\tau_{det} = 1$ ms detection period and assuming a total detection efficiency $\epsilon = 0.3\%$, up to $\epsilon\Gamma\tau_{det}/2 = 10$ photons will be detected per ion that remains in the ground state after clock interrogation. This is sufficient to suppress detection-photon shot noise below the QPN limit. The fraction of excited state ions is used to determine the frequency offset between the clock laser and the atomic transition frequency, and frequency feedback is applied to the clock laser to keep it on resonance.

Fourth, the ions that are in the excited state after clock interrogation are repumped to the ground state by driving the 379 nm $^3P_0 \leftrightarrow {}^1P_1$ transition. The $^1P_1$ state quickly decays to $^1S_0$, with negligible branching ratios for decay to $^3P_1$ and $^3P_2$. Optionally, the total number of ions can be determined by performing readout again after repumping. Subsequently, the next cycle of clock operation can begin.

**Many-ion clock considerations**

Three-dimensional Coulomb crystals of many ions can be confined in a single linear RF Paul trap[14]. For large Coulomb crystals of a fixed number of ions, there are multiple local minima of the total zero-temperature energy consisting of the Coulomb energy due to repulsion of the ions from each other, the time-averaged Coulomb energy of the ions interacting with the trapping fields, and the time-averaged kinetic energy of the micromotion (i.e., the pseudopotential). These local minima correspond to stable geometric crystal configurations, in which each ion is spatially localized and oscillates about its equilibrium position due to micromotion and secular motion. Distinct crystal configurations have different sets of equilibrium ion positions. Different configurations are observed both theoretically[46] and experimentally[16,47] when ions are laser cooled to a crystalline state starting from a non-localized cloud state with identical initial thermal properties, for example after loading the trap or a collision with a background gas molecule, and the number of configurations grows exponentially with the number of ions[48]. We perform molecular dynamics calculations of the ion motion in two steps. First, similar to Refs. 21,24, we calculate the equilibrium ion positions by time-evolving

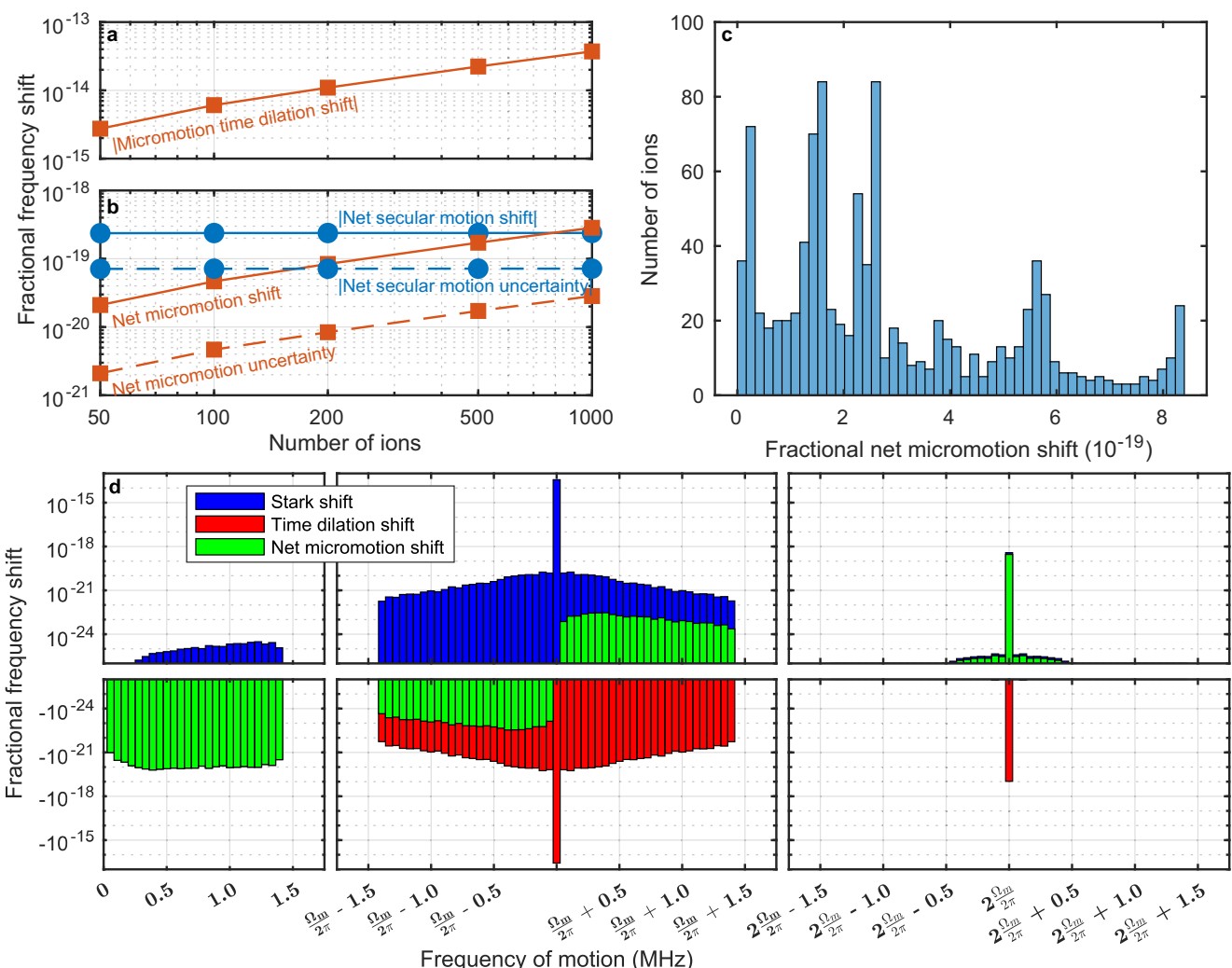

**Fig. 3 | Systematic frequency shifts due to ion motion. a** Absolute value of the time-dilation shift due to micromotion, averaged over all of the ions, as a function of the number of ions in the trap. As the number of ions increases, some of the ions are located further away from the trap axis, leading to an increase in the average micromotion time-dilation shift. **b** Absolute value of the net micromotion and secular motion shifts, averaged over all of the ions, as a function of the number of ions in the trap. The net secular motion (micromotion) shift is defined as the sum of the negative time-dilation shift and the positive Stark shift caused by the electric field of the ion trap due to ion motion at the secular motional frequencies (integer multiples of the trap drive frequency). Also shown are estimated uncertainties in how well these shifts can be characterized. The secular motion temperature here and in panel d is assumed to be 120 $\mu$K. **c** Histogram showing the number of ions out of a 1000 ion Coulomb crystal with a net micromotion shift within each bin. Each ion is located at a different position within the trap and thus experiences different

micromotion and Stark shifts, but the inhomogeneous broadening associated with these shifts is below $10^{-18}$ fractionally and should not impact spectroscopic coherence within the foreseeable future. **d** Histogram of motional frequency shifts as a function of the frequency of the ion motion, for a 1000 ion Coulomb crystal. All 3000 secular modes have frequencies below 1.5 MHz. For motion at these frequencies, the magnitude of the time-dilation shift is much larger than the Stark shift due to trapping fields and the green bars indicating the net shift are covering up the red bars indicating the time-dilation shift. At the magic trap drive frequency $\Omega_m/(2\pi) \approx 225$ MHz, the micromotion time-dilation shift is exactly canceled by the corresponding trap Stark shift, but at secular motion sidebands of the trap drive frequency this cancellation is imperfect. At the second harmonic of the trap drive frequency, the Stark shift is much larger than the time-dilation shift and dominates the net shift.

the equations of motion in the pseudopotential approximation starting from random initial positions and including a viscous damping force that qualitatively represents laser cooling. For each set of trap parameters and number of ions investigated, we generate up to 36 configurations in parallel using a high-performance computer cluster. For ion numbers up to 1000, this takes < 1 week. The strength of the damping force is such that the cooling time constant is roughly 100 $\mu$s, which is a typical experimental cooling time constant, and we have verified that reducing this strength by an order of magnitude does not reduce the number of distinct configurations. Once the equilibrium ion positions are known, it is straightforward to compute the frequencies and eigenvectors of the normal modes of secular motion and their Lamb-Dicke parameters[49,50], and the spectrum of the clock

transition[51]. Second, we calculate the amplitude and direction of the micromotion of each ion and verify the stability of the ion crystal by time evolving the full equations of motion (i.e., without taking the pseudopotential approximation, similar to refs. 52,53), starting from the ion positions computed in the first step.

Figure 3a, b show the fractional frequency shifts due to secular motion, micromotion, and the Stark shift due to the trapping fields, averaged over all of the ions, and estimates of the uncertainties with which these shifts can be characterized. For these calculations, the trap is assumed to be driven at the magic RF frequency with voltages such that the secular frequencies of a single ion in the trap are 1.15 MHz, 0.85 MHz, and 0.10 MHz. The number of ions varies from 50–1000. The magnitude of the average micromotion time-dilation shift

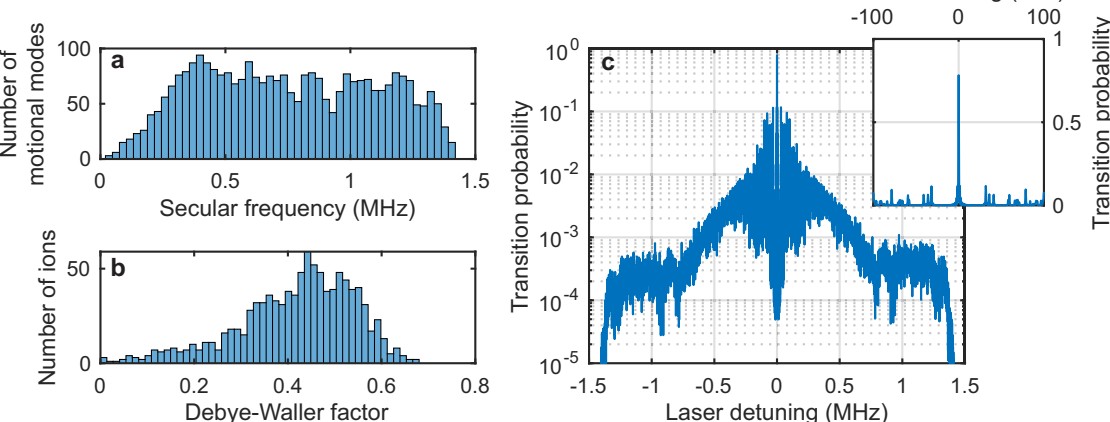

**Fig. 4 | Effects of ion motion in a 1000 ion Coulomb crystal at a motional temperature of 120 $\mu$K on spectroscopy of the clock transition.** **a** Histogram showing the number of collective motional normal modes binned by the motional mode frequency, which ranges from a minimum of 32 kHz up to a maximum of 1.419 MHz. **b** Histogram showing the number of ions binned by the Debye-Waller factor due to secular motion, which relates the Rabi frequency for driving the clock transition in an ion at rest with the Rabi frequency for an ion undergoing secular motion. **c** Calculated spectrum of the clock transition probed using Rabi spectroscopy with a probe duration of 1 ms and a laser intensity that maximizes the on-resonance transition probability, averaged over all of the ions. The inset is an enlargement of the region near the atomic transition frequency, showing the carrier at zero detuning and a few discrete motional sidebands. For larger detunings shown in the main plot, the motional sidebands overlap in frequency to form broad bands.

increases with the number of ions as ions are pushed further away from the trap axis, along which the AC electric field is zero, reaching $3.7 \times 10^{-14}$ for 1000 ions. The time-dilation shift due to micromotion at exactly the magic trap drive frequency is perfectly canceled by the Stark shift due to the electric field that drives the motion, as indicated in Fig. 3d. There is also a time-dilation shift due to micromotion at twice the trap drive frequency, which is not canceled by the corresponding Stark shift, and this motion prevents perfect cancellation of micromotion time dilation[40]. We define the net micromotion shift to be the sum of the time dilation shift and Stark shift caused by the trap electric field in the reference frame of the ion due to ion motion at integer multiples of the trap drive frequency. The net micromotion shift after this imperfect cancellation for 1000 ions is $2.8 \times 10^{-19}$, and it should be possible to characterize this shift to achieve an uncertainty that is at least one order of magnitude smaller. Figure 3c shows a histogram of the net micromotion shifts of each ion individually; the sub-$10^{-18}$ inhomogeneous broadening of this distribution will not limit spectroscopic coherence within the foreseeable future. The net secular motion shift (defined as the sum of the time dilation shift and Stark shift due to ion motion at the secular frequencies) for 1000 ions is $-2.4 \times 10^{-19}$, and we conservatively estimate that this can be characterized with an uncertainty of $7.2 \times 10^{-20}$. Importantly, the differences of these shifts for different ion crystal configurations are smaller than our estimated uncertainties, so $Sn^{2+}$ clocks will be robust to ion loss and background gas collisions that cause the crystal to melt and recrystallize in a different configuration.

Ion motion leads to motional sidebands on the clock transition and an inhomogeneous reduction of the Rabi frequency for driving the carrier transition due to the Debye-Waller factors of each ion[51,54]. Micromotion is dominantly along the radial trap directions and its effect on the clock transition spectrum can be nearly eliminated by probing along the axial trap direction. Figure 4a shows the spectrum of the secular motional mode frequencies for a 1000 ion crystal with the same trap parameters as above. The distribution of Debye-Waller factors of each ion for axial probing due to secular motion at a temperature of 120 $\mu$K, corresponding to the Doppler cooling limit with a 5 MHz power broadened linewidth, is shown in Fig. 4b. The contrast for Rabi spectroscopy of the clock transition is maximized by setting the laser intensity and pulse duration such that ions with the median value of the Rabi frequency undergo a $\pi$-pulse when the laser is on

resonance. Figure 4c shows the spectrum of the clock transition under these conditions. The contrast is limited to 79 % due to the inhomogeneous reduction of the Rabi frequency by the Debye-Waller effect, and a broad band of overlapping motional sideband transitions surrounds the carrier transition which is at zero detuning. The lowest frequency motional sideband transitions are above 14 kHz for any of the generated crystal configurations, and should not lead to significant line-pulling due to off-resonant excitation for probe times longer than about 10 ms.

The trap parameters can be optimized to minimize the net micromotion shift, minimize the size of the ion crystal and thus inhomogeneous broadening due to magnetic field gradients as well as required laser beam diameters, maximize the lowest secular frequency, maximize the spectroscopy contrast, or for other goals. Furthermore, the number of ions can be selected to balance systematic shifts and inhomogeneous broadening that increase with the number of ions versus the QPN stability limit that decreases with increasing numbers of ions. Figure 5 shows a limited subset of this trade space. The lowest normal mode secular frequency for trap parameters such that the single-ion secular frequencies are 1.15 MHz, 0.85 MHz, and 0.10 MHz decreases as the number of ions increases, reaching a mean value of 32 kHz for 1000 ions with a distribution spanning from 15 kHz – 56 kHz for different crystal configurations. For axial single-ion secular frequencies that are too low or too high relative to the radial single-ion secular frequencies, the spectroscopy contrast is reduced. The two radial single-ion secular frequencies must be different in order to avoid a zero-frequency mode in which the ion crystal rotates about the trap axis, and more generally low frequency motional modes should be avoided becuase they have higher heating rates due to electric field noise[55]. A more complete molecular dynamics model incorporating the details of the laser cooling can be used to optimize the cooling parameters and is likely necessary to provide a rigorous constraint on the corresponding Doppler shift uncertainty of the clock. Single-ion secular frequencies of 1.15 MHz, 0.85 MHz, and 0.10 MHz represent a trade-off of the aforementioned optimization goals that enable both high accuracy and high stability clock performance.

These secular frequencies correspond to Mathieu parameters of $|q_{x,y}| = 0.0127$, $a_x = 2.33 \times 10^{-5}$, and $a_y = -2.41 \times 10^{-5}$, and can be achieved by differentially driving a trap with 500 $\mu$m ion-to-electrode

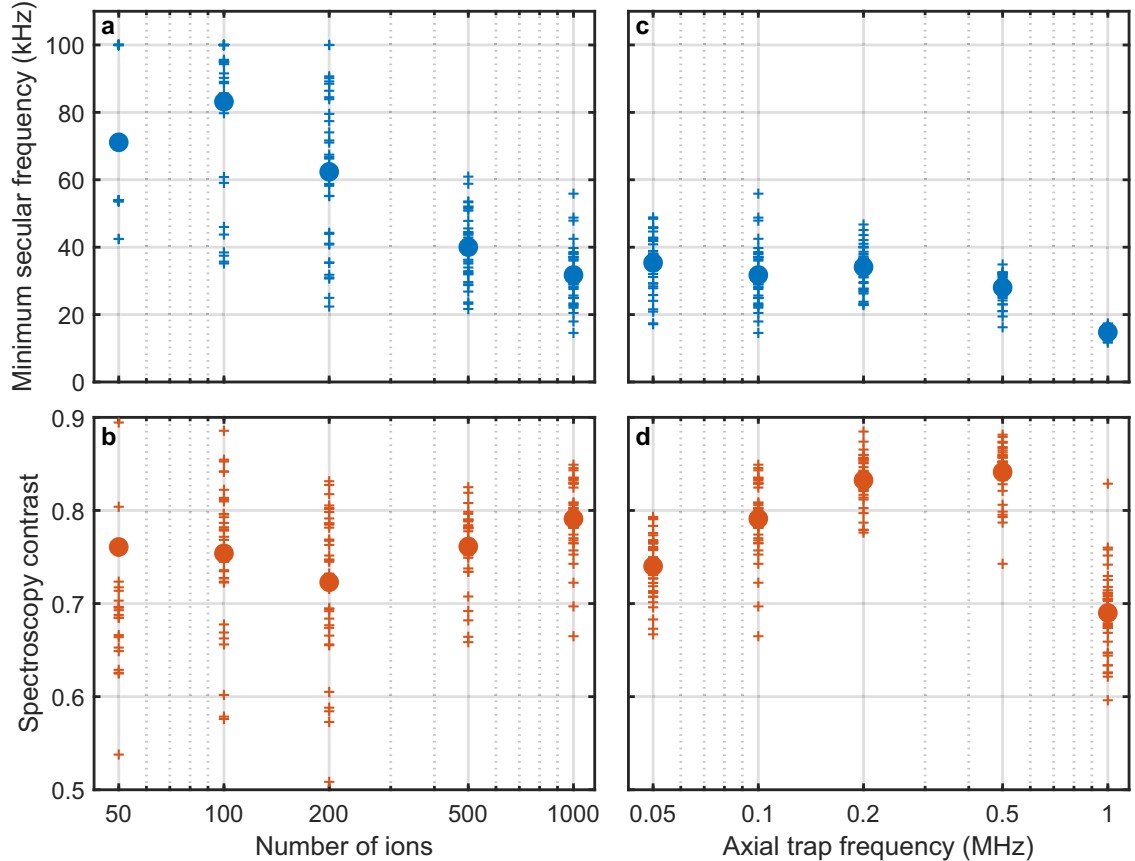

**Fig. 5 | Minimum secular frequency and maximum possible spectroscopic contrast at a motional temperature of 120 $\mu$K for different Coulomb crystal configurations. a** Minimum secular frequency and (**b**) average spectroscopy contrast as a function of the number of trapped ions for trap parameters such that the secular frequencies of a single $Sn^{2+}$ ion are 1.15 MHz, 0.85 MHz, and 0.10 MHz. For Coulomb crystals containing many ions, there are multiple local minima of the total Coulomb energy corresponding to distinct stable geometric crystal configurations. For each number of ions, 36 configurations were generated by numerically integrating the equations of motion including a damping term which qualitatively represents laser cooling starting from randomized initial ion positions. Each of the configurations that were generated is plotted as a small cross, while the

average of all of the generated configurations is plotted as a large circle. As the number of ions increases, the minimum secular frequency decreases but the spectroscopy contrast remains relatively constant. Off-resonant driving of low-frequency motional sideband transitions can lead to detrimental line pulling of the clock transition carrier. **c** Minimum secular frequency and (**d**) average spectroscopy contrast as a function of the single-ion axial secular frequency for 1000 ions and single-ion radial secular frequencies 1.15 MHz and 0.85 MHz. As the single-ion axial secular frequency nears degeneracy with the single-ion radial secular frequencies, the minimum secular frequency and the average spectroscopy contrast go down.

distance with 970 V amplitude of rf. This smaller-than-typical $q$ parameter offers several advantages for Coulomb crystal clocks. Molecular dynamics simulations show that rf heating of large ion crystals, which arises due to the breakdown of the pseudopotential approximation $q \ll 1$, scales roughly like $q^6$ at low temperatures[56]. Care must be taken, however, to minimize anharmonicity of the trap potential as this can lead to additional rf heating[57]. Ion heating due to background gas collisions is also reduced for smaller $q$ parameters[58].

## Discussion

### Performance of a 1000 ion $Sn^{2+}$ clock

We estimate that the total systematic uncertainty of a 1000 ion $Sn^{2+}$ clock operated under the conditions specified above could be suppressed to the $9.0 \times 10^{-20}$ level or beyond (see Table 3). The largest shift at 15 G magnetic field is the quadratic Zeeman shift at $-3.7 \times 10^{-15}$, but this can be controlled and characterized very well. Performing a measurement of the quadratic Zeeman shift at 100 G field with a fractional frequency uncertainty of $10^{-18}$ would determine the atomic coefficient such that the corresponding uncertainty of the shift at 15 G is $10^{-18}/(100\,G/15\,G)^2 = 2.3 \times 10^{-20}$. Characterization of the magnetic field with 10 $\mu$G uncertainty would reduce the associated uncertainty of the quadratic Zeeman shift to $4.9 \times 10^{-21}$. This could be

accomplished in-situ by measuring the splitting of the extreme Zeeman sublevels of the metastable $^3P_2$ state with 84 Hz uncertainty, via two-photon excitation from the ground state using lasers detuned from the 181 nm $^1S_0 \leftrightarrow {}^3P_1$ and 2481 nm $^3P_1 \leftrightarrow {}^3P_2$ transitions. The next-largest shift is the probe laser Stark shift for single-photon spectroscopy, but the uncertainty of this shift can be suppressed to a negligible level using hyper-Ramsey[42] or auto-balanced Ramsey[43] interrogation protocols. The $5.2 \times 10^{-18}$ Stark shift due to blackbody

**Table 3 | Fractional systematic shifts and estimated uncertainties of a 1000 ion $Sn^{2+}$ clock, operating at a magnetic field of 15 G in a room temperature ion trap with a blackbody radiation temperature of 300.0(5) K**

| Effect | Shift ($10^{-20}$) | Uncertainty ($10^{-20}$) |
| --- | --- | --- |
| Secular motion | −24 | 7.2 |
| Blackbody radiation shift | 515 | 3.4 |
| Micromotion | 28 | 2.8 |
| Quadratic Zeeman shift | −366550 | 2.3 |
| Probe laser Stark shift | 19753 | 2.0 |
| Total | −346278 | 9.0 |

radiation (BBR) at 300 K can be characterized with an uncertainty of $3.4 \times 10^{-20}$ by measuring the static differential polarizability following Dubé et al.[40] and the dynamic correction using a few far-IR lasers[25,59] (these measurements can be done with a single ion for simplicity), and characterizing the temperature of the BBR with an uncertainty of 500 mK[60]. Alternatively, the BBR shift can be suppressed to a negligible level by cooling the vacuum chamber and thus the BBR to cryogenic temperatures[61–63]. It may, in fact, be necessary to use a cryogenic vacuum chamber to suppress the uncertainty due to background gas collisions to a negligible level (see Methods). The electric quadrupole shift is negligible because the total electronic angular momentum of both clock states is zero, so it is only sourced by mixing of $^3P_0$ with other states[64]. The largest uncertainty in our estimate is due to secular motion time dilation, and it may be possible to reduce this by cooling to lower temperatures or characterizing the motional temperature better than our conservative assumption of 30%. It may be possible to achieve a smaller secular motion and thus total systematic uncertainty by reducing the number of ions, but this would come at the cost of degraded stability.

The stability of the 1000 ion clock is fundamentally limited by QPN at $4.4 \times 10^{-18}/\tau^{1/2}$ where $\tau$ is the measurement duration in seconds, assuming a conservative 1 s probe duration and 80 % duty cycle. This is nearly two orders of magnitude better than the best clock-comparison stability achieved with an ion clock[6,65] and one order of magnitude better than the best stability comparison of independent lattice clocks[66]. At this level, it is possible to perform frequency measurements with a total uncertainty of $10^{-19}$ in under 3 h, measure the time-dependent gravitational effects of solid-Earth tides with high signal-to-noise ratio[67], and search for ultralight bosonic dark matter (DM) candidates over a broad range of particle masses[68].

It may be possible to achieve yet higher performance than discussed here by co-trapping a second ion species such as Sr$^+$ together with Sn$^{2+}$ in the trap. Due to their higher mass-to-charge ratio, the Sr$^+$ ions experience a weaker radial restoring force from the trap and thus fill the Coulomb crystal lattice sites further from the trap axis, causing the Sn$^{2+}$ ions to reside in a cylindrical volume of lattice sites closer to the trap axis[69] where the micromotion shift is smaller. Sr$^+$ could be used to perform sympathetic laser cooling of the two-species Coulomb crystal[70] both with a higher cooling rate on the 422 nm $S_{1/2} \leftrightarrow P_{1/2}$ transition before clock interrogation and to a lower temperature continuously during clock interrogation using the 674 nm $S_{1/2} \leftrightarrow D_{5/2}$ transition. The magnetic field could be characterized by measuring the splitting of the Zeeman sublevels of the $D_{5/2}$ state. Using these techniques, it may also be possible to increase the ion number beyond 1000 and achieve even higher precision while maintaining sub-$10^{-19}$ inaccuracy.

## Sensitivity to beyond Standard Model physics

Precision measurements of frequency differences between different isotopes of two or more atomic transitions[71–73] can be used to search for hypothetical new bosons in the intermediate mass range that mediate interactions between quarks and leptons[74,75]. Such isotope shift data are analyzed using a King plot, where mass-scaled frequency shifts of two optical transitions are plotted against each other for a series of isotopes[76]. The leading standard model contributions to the isotope shift (IS), mass and field shifts, give a linear relationship between two electronic transitions with respect to different IS measurements. New spin-independent interactions will break this relation, and thus can be probed by looking for a non-linearity of the King plot[74,75]. Data for at least four even isotopes are needed to detect the non-linearity. In addition to the $^1S_0 \leftrightarrow {}^3P_0$ clock transition that we focus on in this article, Sn$^{2+}$ also has a $^1S_0 \leftrightarrow {}^3P_2$ two-photon clock transition, which can be driven with the 181 nm Doppler cooling laser together with a 2481 nm laser, that is capable of supporting very high accuracy and precision spectroscopy.

Higher-order standard model contributions could break the linearity of the King plot as well[75] and must either be calculated with high accuracy[77], which is very difficult, or eliminated together with uncertainties of the isotope mass differences using more transitions and/or isotopes in a generalized analysis[78,79]. In particular, nuclear structure properties such as the quadrupole deformation and higher order nuclear moments can cause a non-linearity in the King plot. Sn$^{2+}$ is unique in that it is the element with the greatest number of stable isotopes: in particular, there are seven stable isotopes with zero nuclear spin ($^{112}$Sn$^{2+}$, $^{114}$Sn$^{2+}$, $^{116}$Sn$^{2+}$, $^{118}$Sn$^{2+}$, $^{120}$Sn$^{2+}$, $^{122}$Sn$^{2+}$, and $^{124}$Sn$^{2+}$), an additional nuclear-spin-zero isotope with a lifetime of 230,000 years ($^{126}$Sn$^{2+}$), and ten more nuclear-spin-zero isotopes with a lifetime > 1 s that could be studied at rare isotope facilities. Additional transitions in other charge states could also be measured: the 583 nm $^3P_0 \leftrightarrow {}^1S_0$ intercombination transition in Sn, the sub-Hz linewidth 2352 nm $^2P_{1/2} \leftrightarrow {}^1P_{3/2}$ magnetic dipole transition in Sn$^+$, and perhaps more. Therefore, Sn$^{2+}$ is particularly well-suited for new physics searches with the generalized analysis[78,79] where nuclear structure uncertainties are removed by using data from more isotopes in order to separate higher order Standard Model effects in the nucleus and the signal due to a hypothetical new boson[72,73]. This is especially important in light of recent experiments in neutral Yb and Yb$^+$ that detected the non-linear effects that may be caused by higher-order contributions within the standard model[71–73].

Hypothesized ultralight bosonic DM candidates would behave as a highly-coherent classical field that oscillates at the Compton frequency corresponding to the DM particle mass $f_\phi = m_\phi c^2/h$, where $m_\phi$ is the mass, and $c$ is the speed of light. This field is predicted to couple to atomic transition frequencies and would be detectable by measuring the ratio of two atomic transition frequencies or the difference between an atomic transition frequency and a Fabry-Perot resonator that have different sensitivities to the DM coupling[68]. Many-ion clocks can be sensitive to a broad range of DM particle masses due to their high stability at short measurement durations discussed above. Although in standard clock operation the upper limit to the measurement bandwidth and thus DM particle mass is given by the reciprocal of the probe duration, dynamical decoupling can be used to extend sensitivity up to much higher frequencies[68]. The high stability of Sn$^{2+}$ clocks also makes them competitive with lattice clocks for proposed space tests of general relativity[80].

## Summary and outlook

We have proposed Coulomb-crystal optical atomic clocks based on many Sn$^{2+}$ ions confined in a linear RF Paul trap as a candidate for a new metrological platform that offers the possibility of accuracy and precision better than any present-day optical lattice or single-ion clock. Sn$^{2+}$ is unique among candidates for Coulomb-crystal clocks because it has a highly forbidden intercombination clock transition between two states with zero electronic angular momentum, a spin-zero nucleus that eliminates nonscalar perturbations, a negative differential polarizability that enables cancellation of the micromotion time-dilation shift by the associated Stark shift, and an accessible direct laser cooling and readout transition. Beyond applications in timekeeping, this platform has the potential to find applications in relativistic geodesy, searches for physics beyond the Standard Model, and more.

## Methods

### Quadrupole moment

The quadrupole moment $\Theta$ of an atomic state $|J\rangle$ is given by

$$\Theta = \langle J, M_J = J | Q_0 | J, M_J = J \rangle = \sqrt{\frac{J(2J-1)}{(2J+3)(J+1)(2J+1)}} \langle J \| Q \| J \rangle, \quad (3)$$

where $\langle J||Q||J\rangle$ is the reduced matrix element of the electric quadrupole operator.

For the $5s5p\,{}^3P_2$ state, we find

$$\langle {}^3P_2||Q||{}^3P_2\rangle \approx -6.9\,ea_0^2 \tag{4}$$

and

$$\Theta({}^3P_2) \approx -1.7\,ea_0^2. \tag{5}$$

## Zeeman shift

Since we consider the ion with $J = I = 0$, there is no linear Zeeman shift. The second-order Zeeman shift, $\Delta E$, is given by[81]

$$\Delta E \equiv \Delta E^{(1)} + \Delta E^{(2)} = -\frac{1}{2}\alpha^{M1}B^2 + \frac{\mu_0^2 m}{2\hbar^2}\sum_{i=1}^{Z}\langle 0|(\mathbf{B}\times\mathbf{r}_i)^2|0\rangle, \tag{6}$$

where $\alpha^{M1}$ is the magnetic-dipole static polarizability, $\mu_0$ is the Bohr magneton, $m$ is the electron mass, and $\mathbf{r}_i$ is the position vector of the $i$th electron. For a state $|J=0\rangle$, the polarizability can be written as

$$\alpha^{M1} = \frac{2}{3}\sum_n \frac{|\langle n||\mu||J=0\rangle|^2}{E_n - E_0}. \tag{7}$$

To estimate the shift for the clock transition due to this term, we note that the $\alpha^{M1}({}^1S_0)$ polarizability is negligibly small compared to $\alpha^{M1}({}^3P_0)$, so we have

$$\Delta\nu^{(1)} \equiv \frac{\Delta E({}^3P_0) - \Delta E({}^1S_0)}{h} \approx \Delta E({}^3P_0)/h.$$

For an estimate of $\alpha^{M1}({}^3P_0)$, we take into account that the main contribution to this polarizability comes from the intermediate state $5s5p\,{}^3P_1$. Then, from Eq. (7), we obtain

$$\alpha^{M1}({}^3P_0) \approx \frac{2}{3}\frac{\langle {}^3P_1||\mu||{}^3P_0\rangle^2}{E_{{}^3P_1} - E_{{}^3P_0}}. \tag{8}$$

We found the matrix element $|\langle {}^3P_1||\mu||{}^3P_0\rangle|$ to be $1.4064\,\mu_0$ and $1.4053\,\mu_0$ in the CI+all-order and CI+MBPT approximations, respectively. These values differ by 0.07 %. According to our calculation, the next (after ${}^3P_1$) intermediate state in Eq. (7), ${}^1P_1$, contributes to $\alpha^{M1}({}^3P_0)$ at the level of 0.06 % compared to the contribution of the ${}^3P_1$ state. Conservatively, estimating the contribution of all other intermediate states as $2\times 0.06$ %, we arrive at the uncertainty of $\Delta\nu^{(1)}$ at the level of 0.2 %.

Using it and the experimental value of the energy difference $E_{{}^3P_1} - E_{{}^3P_0} \approx 1648$ cm$^{-1}$, we arrive at

$$\Delta\nu^{(1)} = -26.16(5)\,\frac{\text{mHz}}{\text{G}^2}\,B^2,$$

where the magnetic field $B$ is expressed in G.

One can show[81] that the contribution of the second term in Eq. (6) to the second-order Zeeman shift of the clock transition can be written as

$$\Delta\nu^{(2)} \equiv \frac{(2\pi\mu_0)^2 m}{3h^3}B^2 \times \sum_{i=1}^{Z}\{\langle {}^3P_0|r_i^2|{}^3P_0\rangle - \langle {}^1S_0|r_i^2|{}^1S_0\rangle\}, \tag{9}$$

where the matrix elements are expressed in $a_0^2$.

The calculation, carried out in the framework of our approach, leads to

$$\sum_{i=1}^{Z}\{\langle {}^3P_0|r_i^2|{}^3P_0\rangle - \langle {}^1S_0|r_i^2|{}^1S_0\rangle\} \approx 1.7\,a_0^2. \tag{10}$$

To check this result, we made another calculation, reconstructing the basis set and using the pure CI method, where all 48 electrons were placed in the valence field, and several of the most important configurations were included into consideration. We obtained the same value $1.7\,a_0^2$.

Using this result, we find

$$\Delta\nu^{(2)} = 0.17(2)\,\frac{\text{mHz}}{\text{G}^2}\,B^2.$$

We estimate the uncertainty of this term at the level of 10 %.

Finally, we find the quadratic Zeeman shift as

$$|\Delta\nu| = |\Delta\nu^{(1)} + \Delta\nu^{(2)}| = 25.98(5)\,\frac{\text{mHz}}{\text{G}^2}\,B^2. \tag{11}$$

## Rabi frequency

The Rabi frequency of the one-photon ${}^1S_0 - {}^3P_0$ transition is determined as

$$\Omega = \left|\left\langle \widetilde{{}^3P_0}|\mathbf{d}\cdot\mathcal{E}|{}^1S_0\right\rangle\right|/\hbar, \tag{12}$$

where $\mathcal{E}$ is the electric field of the laser wave and we designate

$$\left|\widetilde{{}^3P_0}\right\rangle \equiv |{}^3P_0\rangle - \sum_n \frac{|n\rangle\langle n|\boldsymbol{\mu}\cdot\mathbf{B}|{}^3P_0\rangle}{E_{{}^3P_0} - E_n}. \tag{13}$$

Substituting Eq. (13) to Eq. (12), we arrive after simple transformation at

$$\hbar\Omega = \frac{\mathbf{B}\cdot\mathcal{E}}{3}\left|\sum_n \frac{\langle {}^3P_0||\mu||n\rangle\langle n||d||{}^1S_0\rangle}{E_n - E_{{}^3P_0}}\right|. \tag{14}$$

Leaving (for an estimate) in the sum of Eq. (14) only two intermediate states, $5s5p\,{}^3P_1$ and $5s5p\,{}^1P_1$, and using our calculated values $|\langle {}^3P_0||\mu||{}^3P_1\rangle| \approx 1.406\,\mu_0$ and $|\langle {}^3P_1||d||{}^1S_0\rangle| \approx 0.249\,ea_0$, we find

$$|\Omega|/(2\pi) \approx 3.4\,\frac{\text{mHz}}{\text{G}\sqrt{\text{mW/cm}^2}}\,\sqrt{I}B\cos\theta, \tag{15}$$

where $I$ is the laser intensity and $\theta$ is the angle between $\mathbf{B}$ and $\mathcal{E}$.

The clock transition can also be driven as the two-photon $E1 + M1$ transition. The respective formalism was developed in Ref. 23. The $E1 + M1$ Rabi frequency, $\Omega_2$, is given by

$$\Omega_2 = \frac{4\pi}{3\varepsilon_0 hc^2}\sqrt{I_1 I_2}\,\Lambda, \tag{16}$$

where $\Lambda$ can be written in terms of dominant contributions as

$$\Lambda \approx \left|\sum_{n={}^3P_1,{}^1P_1}\frac{\langle {}^3P_0||\mu||n\rangle\langle n||d||{}^1S_0\rangle}{E_n - E_{{}^3P_0}/2}\right| \tag{17}$$

and $I_1$ and $I_2$ are the intensities of two probe laser waves. Using the experimental energies and the values of the matrix elements given in

Table 1, we find

$$\frac{\Omega_2}{2\pi} \approx 0.27 \frac{\text{Hz}}{\text{kW/cm}^2} \sqrt{I_1 I_2}. \tag{18}$$

**Second-order Zeeman shift due to the ac magnetic field at the trap drive frequency**

We estimate the size of this effect with the following simple model. We assume that the ion trap has a 500 $\mu$m ion-to-electrode distance and is driven differentially with an RF voltage amplitude of 970 V, as is necessary to achieve the secular frequencies proposed here. Standard linear RF Paul traps constructed with cylindrical rods or blade electrodes that are electrically driven from one end have current running down the electrodes due which is sunk by a capacitance at the far ends of the electrodes of order 100 fF. At the center of the trap, symmetry dictates that the RF magnetic field should be zero. At the edge of the ion crystal, 22 $\mu$m away from the trap center, this model predicts that there is an AC magnetic field of 50 mG, which causes an inhomogeneous frequency shift of 4 mHz. It should be possible to characterize the average over all ions of this shift sufficiently well as described in the text, and this is compatible with probe durations up to about 100 s. To achieve longer probe durations it will be necessary to design traps with less capacitance or more symmetric current distributions.

**Background gas collision shift**

A detailed analysis of the background gas collision shift and uncertainty for many ion clocks requires computationally intensive Monte Carlo simulations and is beyond the scope of this article, but in the following we provide a simple estimate based on scaling an analysis of the uncertainty for single-ion clocks[82]. The total rate of Langevin collisions between a Coulomb crystal and background gas molecules scales linearly with the number of ions and linearly with the charge of the ions, so for a 1000 ion $Sn^{2+}$ clock the background gas collision rate will be ~2000 times larger than in single-ion clocks. As a conservative upper bound, we suppose that the systematic uncertainty due to background gas collisions scales linearly with the collision rate. A more detailed model would take into account that only the Doppler shift component of the uncertainty scales with the total collision rate while the phase shift component of the uncertainty (which dominated the collision shift uncertainty in Ref. [82]) scales with the collision rate per ion. Current state-of-the-art single-ion clocks operating in room temperature ultrahigh vacuum (UHV) chambers with background gas pressures of roughly $10^{-10}$ torr have systematic uncertainties due to background gas collisions of order $10^{-19}$ [59,83]. Although according to this conservative estimate background gas collisions may contribute of order $10^{-18}$ uncertainty for 1000 ion $Sn^{2+}$ clocks in room temperature UHV chambers with pressures of at best $10^{-12}$ torr, Monte Carlo simulations will likely be able to constrain the uncertainty to be much smaller.

Alternatively, pressures below $10^{-16}$ torr have been achieved for ion traps operated in cryogenic vacuum chambers[84]. An even more conservative upper bound can be obtained by considering the rate of collisions as follows. For $10^{-16}$ torr of He background gas at 10 K, the total rate of Langevin collisions with any of the ions in a 1000 $Sn^{2+}$ ion crystal is $1 \times 10^{-4}$/s. For a 1 s probe time, this amounts to one collision every $10^4$ probes. At most, this could result in a fractional shift of the measured transition frequency of $10^{-4}$ relative to the spectroscopic linewidth, which in this case is 1 Hz. Thus, the background gas collision shift can be straightforwardly upper bounded to be below $10^{-19}$ for $Sn^{2+}$ clocks operating in cryogenic vacuum chambers. In practice, collisions that deposit a significant amount of kinetic energy to the ion crystal will not contribute to the clock spectroscopy signal due to the Debye-Waller reduction of the Rabi frequency[82], so the actual shift and its uncertainty will be much smaller.

## Data availability

Source data are provided with this paper. All data generated during this study are available from the corresponding author upon request.

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

## Acknowledgements

We thank Kyle Beloy, John J. Bollinger, and David B. Hume and for their careful reading and feedback on this manuscript. This work was supported by the National Institute of Standards and Technology (D.R.L.), the Defense Advanced Research Projects Agency (Atomic-Photonic Integration Program, D.R.L.), the National Science Foundation Q-SEnSE Quantum Leap Challenge Institute (Grant Number OMA-2016244, D.R.L. and M.S.S.), the National Science Foundation (Grant Numbers PHY-2012068 and PHY-2309254, M.S.S. and C.C.), the Office of Naval Research (Grant Numbers N00014-18-1-2634, D.R.L., and N00014-20-1-2513, M.S.S.), and the European Research Council (ERC) under the European Union's Horizon 2020 research and innovation program (Grant Number 856415, M.S.S. and S.G.P.). This research was supported in part through the use of University of Delaware HPC Caviness and DARWIN computing systems: DARWIN - A Resource for Computational and Data-intensive Research at the University of Delaware and in the Delaware Region, Rudolf Eigenmann, Benjamin E. Bagozzi, Arthi Jayaraman, William Totten, and Cathy H. Wu, University of Delaware, 2021, URL: https://udspace.udel.edu/handle/19716/29071. The views, opinions, and/or findings expressed are those of the authors and should not be interpreted as representing the official views or policies of the Department of Defense or the U.S. Government.

## Author contributions

D.R.L. conceived the idea. S.G.P., C.C., and M.S.S. performed the calculations of Sn$^{2+}$ atomic properties. D.R.L. performed the calculations of ion motion in Coulomb crystals. All authors discussed the results and implications and contributed to writing and editing the paper.

## Competing interests

The authors declare no competing interests.
