## [Peer Review File · Nature Communications]

Prospects of a thousand-ion Sn^{2+} Coulomb-crystal clock with sub- 10^{-19} inaccuracyReviewer #1 (Remarks to the Author):

In this manuscript, the authors have proposed Sn^{2+} ion as a new candidate for multi-ions clock. The Sn^{2+} ion has many advantages suitable for multi-ions clock operation, for example, the total electronic angular momentum of both clock states is zero ($J=0$), makes it immune to electric quadrupole shift. The differential static polarizability $\Delta\alpha < 0$, there is a "magic" trap drive frequency Ω_m , in which the time-dilation shift and the Stark shift cancel each other. In the manuscript, the authors give the detailed analysis of Sn^{2+} atomic properties, like energy level, lifetime, magnetic field, second-order Zeeman shift and uncertainty, Stark shift and uncertainty, et al.. And also analyzed the net micromotion shift at "magic" trapping frequency, secular motion shift, blackbody radiation shift. Prospects of a thousand-ion Sn^{2+} Coulomb-crystal clock with sub- 10^{-19} inaccuracy. Although this manuscript only gives the proposal and the calculation, I think the Sn^{2+} ion can be very promising for building an advanced ion optical clock in the near future, with the development of techniques for highly charged ion related experiments, this makes the ion optical clock competitive to the optical lattice clocks on frequency stabilities, thus I think the manuscript reaches the criterion of nature communication, in the aspect of innovation in physics. I recommend the publication of this manuscript in nature communication.

Besides, I have a few comments that need to be explained:

1. Highly charged ion (HCI) candidates for optical clocks are recently a hot topic. However, it seems to be very difficult to make an HCI-based clock until the improvements on both the experimental techniques and the theoretical calculations. I think the authors should mention the recent progress on HCI-based experiments in the instruction, eg. PRA 103, L040801 (2021), PRA 103, 022804 (2021), PRX 11, 041049 (2021) , arXiv:2207.10926.
2. In this manuscript, the author proposed using 1000 Sn^{2+} ion as multi-ion clock. Except the consideration of secular motion, blackbody radiation shift, micromotion, quadratic Zeeman shift, probe laser Stark shift, the author also should consider the background gas collision, collisions between ions and ions lose rate, and give their shifts and uncertainties.
3. How can one make sure the loaded Sn^{2+} ions are pure, and avoid dark ions like Sn^+ co-trapped? Because the dark ions also bring huge heating during clock interrogation and affect micromotion and secular motion.
4. Considering of this large crystal size, the author should not only care about the magnetic field fluctuation but also care about magnetic field gradient. Further, when increases the number of ions, the ions are pushed further away from the trap axis, the RF trap frequency maybe cause large AC magnetic field. The author should give the shifts and uncertainty budget of magnetic field gradient and AC magnetic field.
5. Fig2(c) and line 266: When discussing the uncertainty of the probe laser Stark shift, the uncertainty is a few orders of magnitude smaller than the shift, how is this achieved? By measuring the shift by comparing the transition frequency between the increased power situation and the normal power situation? Or by hyper Ramsey scheme?
6. Line 221: As far as I know, the Doppler cooling efficiency is very sensitive to the power of the cooling laser, for such a large ion crystal, the laser power on each every ion should be different, thus the saturation parameter is not always 20, I don't know if the authors have already considered this in their calculations.
7. Line 275: I'm not sure if I have some misunderstanding on this: for single photon approach, the experiment requires a magnetic field of 15 G, and have a field inhomogeneity of 1 mG over the size of ~ 1 mm. It seems to me that this is very difficult to be achieved by simply adding a pair of Helmholtz

coils.

8. Line 428: As mentioned earlier, I think it is very difficult to be achieved for making a very strong magnetic field with a very small gradient like 1 mG over the size of ~ 1 mm, 10 micro-G is even more difficult.

9. Line 460: "assuming a conservative 1 s probe duration and 80 % duty cycle." Although the lifetime of $3P0$ clock state is quite long ~ 106 s at 15 G magnetic field, ideally one can get very long coherence time, but actually the coherence time is determined by high performance of clock laser and ion heating during interrogation time. What is the evidence for 1 s, is it based on the experience of the Al^+ ion clock or have you done the calculation and analysis? It seems to me that the probe time can be even longer.

10. Line 472: "It may be possible...maintaining sub- 10^{19} inaccuracy." The author proposal using Sr^+ ion to co-trapping Sn^{2+} ions, the outside trapped Sr^+ can push the Sn^{2+} ions towards the trap axis and make the micromotion shift smaller. The issue is, it seems not have a "magic wavelength", cooling Sr^+ ion during clock interrogation can lead into ac Stark shifts. Is the cooling laser off when probing? If so, the authors should consider the heating effect.

Reviewer #2 (Remarks to the Author):

dear editor,

the paper proposed by Leibbrandt et al is a proposal for an optical atomic clock that could be operated with a cloud of one thousand ions as a base, whereas so far, trapped ion based optical clock are operated with a single ion. The idea is to make a more stable clock by interrogation of a thousand ions at the same time and then reduce the quantum projection noise which is the limiting factor for short term stability. The issue is to do that without degrading the accuracy and precision of the clock as the inhomogeneity of the ion environment can induce non uniform shifts responsible for a line broadening and an uncertainty in the shift estimation.

This idea is not new and several ions have been identified that share some characteristics that make them good candidates for this kind of set-up. The main figure of merit is the differential polarizability of the transition. If it is negative, two major effects cancel each-other, provided the RF-trap is operated at a proper frequency, which depends on this difference. This is covered by ref 17-21. The new results of this paper is to add an ion to this list : Sn^{2+} , which have some technical advantage over recent proposals, like the accessibility of the transition wavelength, but requires a strong magnetic field to mix different atomic fine-structure states and make the clock transition accessible to laser excitation. The best performances are reached by a compromise between the lowest possible magnetic field to control Zeeman shifts and the lowest possible laser power to control light induced shifts. If an experiment is able to follow these prescriptions without excess inhomogeneity, the expected clock performance are very good compared to the state of art in terms of uncertainty and inaccuracy. At this stage of the paper, the reader still do not know what isotope is considered. We can guess that an even isotope is assumed but nothing is said about their abundance and possible choices. The calculations used for atomic structure calculations is now well established and one of the author is a reference in the optical clock community for the relevance of her methods.

The discussion regarding the clock operation in the context of a thousand trapped ion is far less convincing. There are a lot of statements regarding the cooling efficiency that look like speculations and are not supported by comparison with published and cited work. If the statements are based on molecular dynamics simulations, it is not written clearly and the fact that 3 laser beams with different

directions are required is not justified anywhere. Indeed, as far as I know, no 3D Coulomb crystal experiments use this set-up as the micro-motion induced Doppler effect make Doppler cooling less efficient (see url = {<https://link.aps.org/doi/10.1103/PhysRevA.105.023101>} for the 2D case where different laser detunings are needed). A single one propagating along the axis is often the best solution for large 3D system. Other estimations lack an established support like the requirement for the magnetic field homogeneity. It is assumed to be better than 0.01% over a mm scale. It should be supported by results coming from similar experiments proving the feasibility of this requirement. A lot of discussion regarding the cloud shape and ion dynamics assume operating parameters that are justified a lot later without any notice. This is very detrimental. The RF optimum frequency is so large and the chosen single-ion secular frequencies are such that they correspond to very low Mathieu parameter, of the order of 10^{-2} . This is at the edge of the stability diagram for an RF-trap and to make it convincing, this proposal should cite experimental results where such a low parameter has been operated with success. If these works do not exist yet, it is because it is really a challenge and this should not be considered as obvious in the article.

The part concerning the different spatial configuration of an ion ensemble is very confusing as the cited works all set the ions in a static pseudo-potential (I suggest the authors to read also Calvo and Yurtsever, EPJD (2007) DOI: 10.1140/epjd/e2007-00137-2), and the authors also use the same strategy, so why mentioning time-averaged kinetic energy of the micro-motion? This is the source of the pseudo-potential and it has nothing to do in a pseudo-potential description, it is one or the other, not both. How micro-motion is obtained in the pseudo-potential approximation? This point deserve clarification. The description of the forces taken into account should be more rigorous and corrected. Indeed, more details about the numerical simulations should be given (like in a methods part) to appreciate the relevance of the results and more earlier works should be cited to support the chosen strategy (see DOI: 10.1103/PhysRevA.82.033406 and <https://doi.org/10.1103/PhysRevLett.70.818> to start with). What about the choice of the damping constant simulating laser cooling? If it is too strong, the system is frozen in a non-equilibrium state. How the authors proceed to make sure this is not the case and that the motional temperature is 120 microK (cf caption of fig 4) ?

The paragraph starting at line 393 is completely speculative as nothing proves that the listed requirements are compatible with each other. It is surprising to read that "maximize the lowest secular frequency" is in the list whereas a so small q_x parameter has been chosen. So this is certainly not the case. Line 413 mention a trade off to reach so small values, so the most important conditions should be enlightened and this need for a maximization should be justified.

On the step of line 458, the requirements concerning the stability and the precision/accuracy of the proposed clock should be clearly identified and separated as some technical choices can be good for one figure of merit but bad for another.

From line 472, another list of speculations appear, concerning the best side ion to be trapped with Sn^{2+} , to compress the ion cloud around the trap axis. The obvious solution would be to increase the cloud density, but once again, the reason why such a low Mathieu parameter is chosen is not given. Furthermore, this idea of using an outer cloud shell to compress the inner cloud is not obvious at all and is not supported by any calculations or previous work. This naive view seems in contradiction with the observation reported in PRL.86.1994. As for the idea of using a multipole trap, I'd like to point to the authors that ref 54 and 56 concern room temperature large ion clouds and in the cold limit, the self organization of ions is very different (see for example J. Phys. B: At. Mol. Opt. Phys. 42 (2009) 154002). For smaller sample, the use of multipole trap is described here : DOI: 10.1103/PhysRevA.81.043410 .

Contrary to what is stated on line 489-494, ions will not gather closer to the axis, because of Coulomb repulsion. (read again section V of ref 17) Anyway, and unfortunately, using a multipole trap with laser cooled ions may be a wrong good idea as stated in this series of paper : <https://doi.org/10.1080/09500340.2017.1408866> and M Marchenay et al 2021 Quantum Sci. Technol. 6 024016 .

At this stage, I think the reader is kind of lost with the come back of spectroscopic considerations from line 495. This would be better placed at the beginning of the paper or it could be suppressed as nothing new is brought here.

As a conclusion, I think this paper is not ready yet for publication. Furthermore, I also think that the presented results are not noteworthy neither original enough for publication by nature communication.

Reviewer #3 (Remarks to the Author):

The manuscript "Prospects of a thousand-ion $\text{Sn}^{\{2+\}}$ Coulomb-crystal clock with sub- $10^{\{-19\}}$ inaccuracy", gives a detailed account on the possibility of building a highly accurate atomic clock based on a large three dimensional ion Coulomb crystal. By considering a clock based on up to 1000 ions, the authors' goal is to produce an atomic clock based on ions, which not only can compete with neutral atom lattice clocks with respect to total achievable inaccuracy, but also with respect to needed experimental integration time. The authors have made a great effort to assess many of the potential limiting issues related to the specific ion and trapping configuration in mind. The manuscript is organized in a logical way by first describing the reason to consider the $\text{Sn}^{\{2+\}}$ ion species and the expected properties of the relevant electronic levels, followed by presenting how the trapping fields and properties of and ion motion within a Coulomb crystal influence the accuracy of the Coulomb crystal clock.

Finally, before a summary and outlook, a discussion on various limiting factors and potential further improvements are briefly presented.

All in all, after a revision of the manuscript with respect to the comments and suggestions stated below, I find the manuscript suitable for publication in Nature Communication, since the work represents a promising scenario for a future new improved time standard.

Comments/questions:

1) In Fig. 1, a Coulomb crystal of 1000 ions is presented. It is obvious that a very elongated crystal along the rf field-free axis is preferable, but it is less clear why trap parameters giving rise to a relatively large difference in the crystal dimensions in the perpendicular plane is chosen. Perhaps the authors could elaborate a bit on that.

2) In Fig. 1, the colors of the clock and cooling beams are very close to each other. Choosing a different color for the cooling beams would be helpful also for the readability of the text on the illustration.

3) Since the "magic" rf frequency of 225 MHz is rather high, a reference that shows this should be achievable with the trap dimensions in mind would be needed.

4) In fig. 2.c, the lines indicating the uncertainties of the second order Zeeman shifts and Stark uncertainties translates into a magnetic and laser field instabilities of $10^{\{-5\}}$ and $10^{\{-4\}}$, respectively. It would be needed with some references that such instabilities should be achievable over the total volume of the Coulomb crystal.

5) With respect to the assessment of time dilation and rf field induced Stark shifts, the authors have considered 36 stationary configurations of the Coulomb crystals which seems reasonable. A single background gas collisions may though lead to dynamical changes in the crystal structure or even lead to crystal melting. Only very few experimental investigations have been made so far to address these issues quantitatively, but they could potentially constitute limitations to use larger crystals in particularly in room temperature trapping environments. Please comment on this in a revised manuscript.

6) With respect to computing the frequencies and eigenvectors of the normal modes of the secular motion and Lamb-Dicke parameters reference is given to [41] in the manuscript. Since this paper deals only with linear chains of ions, a more suitable reference for the current proposal could, e.g., be New J. Phys. 14 093023 (2012).

7)

Fig. 3, which is central to the manuscript, contains a lot of information presented in a condensation fashion. For the non-expert it can be very hard to be sure to have understood it fully as it is.

In fig. 3.b, it would be great to have a clearer physical description of what is meant by “net secular motion shift”. As it looks like it is independent of the number of ions, I assume it accounts only for the time dilation related to the secular motion? Could there also be a secular motion dependent Stark shift which should be accounted for?

Fig. 3.d is very challenging. If I am right that the green bars are just the difference between the red and blue, I believe one could communicate that clearer. The two first panels deal with the secular motion, while the rest of the panels deal with rf induced micromotion. For the sake of clarity, it would be good to separate Fig. 3.d. into three separate sub-figures (d.,e., and f.) and not for all panels use the term “net micromotion shift”.

In the last four panels the fractional frequency shifts exhibit large increases at the magical rf frequency. Why is this the case? This should be explained more carefully in the text. The assumed temperature for the secular contributions should be stated in the figure caption.

8) Somewhere in the manuscript it would be great to have a short discussion of what would the influence of a radial temperature gradient in the crystal. A priori one cannot expect to be able to cool all ions in the crystal to the same final temperature.

9) With respect to uncertainty in the shifts due to BBR, is an effective temperature with an uncertainty of 500 mK reasonable over the whole crystal lengths (~ 1 mm)?

10) With respect to the two species Coulomb crystal discussion on p. 8 and 9, it should be noted that volume of the Sn^{2+} ions in this case will be a nearly perfect cylinder, and hence for a given crystal length and number of ions the outermost ions will be positioned closer to the trap axis when compared to the case of a single Sn^{2+} species ellipsoidal crystal. Consequently, this would lead to a smaller average time dilation and Stark shifts. A potential reference in this relation could, e.g., be Phys. Rev. Lett. 86, 1994 (2001).

Response to referee comments on manuscript “Prospects of a thousand-ion Sn²⁺ Coulomb-crystal clock with sub-10⁻¹⁹ inaccuracy”

David R. Leibrandt,^{1,2,3,*} Sergey G. Porsev,⁴ Charles Cheung,⁴ and Marianna S. Safronova⁴

¹*Department of Physics and Astronomy, University of California, Los Angeles, California, 90095, USA*

²*Time and Frequency Division, National Institute of Standards and Technology, Boulder, Colorado, 80305, USA*

³*Department of Physics, University of Colorado, Boulder, Colorado, 80309, USA*

⁴*Department of Physics and Astronomy, University of Delaware, Newark, Delaware 19716, USA*

(Dated: July 17, 2023)

I. INTRODUCTION

We thank the editor and referees for their thorough and insightful comments, which have helped us to improve the manuscript considerably.

Furthermore, we apologize for the unusually long delay between receiving the referee’s comments and this reply. One of us (Leibrandt) has moved to a new position (from NIST in Boulder, CO to UCLA in Los Angeles, CA) and during this period of time was not able to devote sufficient time to address the referee’s comments until now.

In the following, we reply to the referee’s comments point by point. The referee comments are reproduced verbatim in gray text, followed by our responses in black text. We have appended a copy of the revised manuscript with the changes highlighted.

II. REVIEWER # 1

In this manuscript, the authors have proposed Sn²⁺ ion as a new candidate for multi-ions clock. The Sn²⁺ ion has many advantages suitable for multi-ions clock operation, for example, the total electronic angular momentum of both clock states is zero (J=0), makes it immune to electric quadrupole shift. The differential static polarizability $\Delta\alpha < 0$, there is a “magic” trap drive frequency Ω_m , in which the time-dilation shift and the Stark shift cancel each other. In the manuscript, the authors give the detailed analysis of Sn²⁺ atomic properties, like energy level, lifetime, magnetic field, second-order Zeeman shift and uncertainty, stark shift and uncertainty, et al.. And also analyzed the net micromotion shift at “magic” trapping frequency, secular motion shift, blackbody radiation shift. Prospects of a thousand-ion Sn²⁺ Coulomb-crystal clock with sub-10-19 inaccuracy. Although this manuscript only gives the proposal and the calculation, I think the Sn²⁺ ion can be very promising for building an advanced ion optical clock in the near future, with the development of techniques for highly charged ion related experiments, this makes the ion optical clock competitive to the optical lattice clocks on frequency stabilities, thus I think the manuscript reaches the criterion of nature communication, in the aspect of innovation in physics. I recommend the publication of this manuscript in nature communication.

We thank the reviewer for their constructive feedback and for supporting the publication of our manuscript in Nature Communications.

Besides, I have a few comments that need to be explained:

1. Highly charged ion (HCI) candidates for optical clocks are recently a hot topic. However, it seems to be very difficult to make an HCI-based clock until the improvements on both the experimental techniques and the theoretical calculations. I think the authors should mention the recent progress on HCI-based experiments in the instruction, eg. PRA 103, L040801 (2021), PRA 103, 022804 (2021), PRX 11, 041049 (2021) , arXiv:2207.10926.

Yes, there has been significant progress on HCI clocks in the past few years with the demonstration of the Ar¹³⁺ clock, and some of the experimental techniques will also be useful for Sn²⁺ clocks. We have added the sentence “Finally, recent demonstrations have shown that it is possible to build clocks based on highly charged ions which can have lower sensitivity to environmental perturbations and higher sensitivity to physics beyond the Standard Model [1–4].”

* leibrandt@physics.ucla.edu

2. In this manuscript, the author proposed using 1000 Sn²⁺ ion as multi-ion clock. Except the consideration of secular motion, blackbody radiation shift, micromotion, quadratic Zeeman shift, probe laser Stark shift, the author also should consider the background gas collision, collisions between ions and ions lose rate, and give their shifts and uncertainties.

We have added the following paragraph to the methods:

“A detailed analysis of the background gas collision shift and uncertainty for many ion clocks is beyond the scope of this article, but in the following we provide a simple estimate based on scaling an analysis of the shift for single-ion clocks [5]. The total rate of Langevin collisions between a Coulomb crystal and background gas molecules scales linearly with the number of ions and quadratically with the charge of the ions, so for a 1000 ion Sn²⁺ clock the background gas collision rate will be approximately 4000 times larger than in single-ion clocks. As a conservative upper bound, we suppose that the systematic uncertainty due to background gas collisions scales linearly with the collision rate. A more detailed model would take into account that only the Doppler shift component of the uncertainty scales with the total collision rate while the phase shift component of the uncertainty scales with the collision rate per ion. Current state-of-the-art single-ion clocks operating in room temperature ultrahigh vacuum (UHV) chambers with background gas pressures of roughly 10⁻¹⁰ torr have systematic uncertainties due to background gas collisions of order 10⁻¹⁹ [6, 7]. Although background gas collisions would likely contribute significant uncertainty for 1000 ion Sn²⁺ clocks in room temperature UHV chambers, pressures below 10⁻¹⁶ torr have been achieved in cryogenic vacuum chambers [8]. Thus, by operating Sn²⁺ clocks in cryogenic vacuum chambers it should be possible to suppress the collision shift uncertainty to below 10⁻²¹.”

As ions are lost from the trap or if they undergo chemical reactions to become dark ions, the crystal configuration will change. However, one important result of our simulations is that the systematic shifts do not change at a level that would increase the overall uncertainty of the clock even as the ion crystal configuration changes. We have added the following sentence to the text to highlight this feature: “Importantly, the differences of these shifts for different ion crystal configurations are smaller than our estimated uncertainties, so Sn²⁺ clocks will be robust to ion loss and background gas collisions that cause the crystal to melt and recrystallize in a different configuration.”

3. How can one make sure the loaded Sn²⁺ ions are pure, and avoid dark ions like Sn⁺ co-trapped? Because the dark ions also bring huge heating during clock interrogation and affect micromotion and secular motion.

During ion loading, the ion trap can be operated in a way such that Sn²⁺ is stably trapped but Sn⁺ is not trapped, either by setting the Mathieu parameters to make Sn⁺ unstable as in a residual gas analyzer or by applying an oscillating electric field resonant with the motional frequency of uncrystallized Sn⁺ to heat it out of the trap. These techniques to kick undesired ions out of the trap become more difficult for impurity ions with a charge-to-mass ratio very close to the desired isotope of Sn²⁺, such as undesired isotopes of Sn²⁺. For these, however, it is possible to use a laser that is blue detuned with respect to the ¹S₀ ↔ ³P₁ transition in the undesired isotope to heat the ions out of the trap. This laser would be far detuned from the ¹S₀ ↔ ³P₁ transition of the desired isotope and would not cause significant heating of the desired isotope.

Despite use of these or other methods, the ion crystal may still contain a small fraction of impurity ions. However, these impurity ions will not significantly affect the clock performance. Impurity ions will change the spectrum of the secular motional modes [9], but all of the mode frequencies will still be within the Doppler cooling bandwidth, so the Doppler shift and uncertainty will be unchanged. The micromotion of each individual ion in a mixed species ion crystal is primarily determined by the charge-to-mass ratio of that ion, so the micromotion Doppler shift and uncertainty of the clock ions will not be significantly changed by impurity ions.

In regard to the heating comment, we assume that the comment refers to RF heating, which is large when ion motion is a significant fraction of the distance between ions and the ions experience the anharmonicity of the potential due to neighboring ions. This heating needs to be overcome with a large laser cooling rate in order to crystallize initially hot and decrystallized ions [10], but is negligible for ion crystals near the Doppler temperature limit [11], even for mixed species ion crystals where only one species is laser cooled [12]. Thus, heating due to impurity ions during clock operation will be negligible.

4. Considering of this large crystal size, the author should not only care about the magnetic field fluctuation but also care about magnetic field gradient. Further, when increases the number of ions, the ions are pushed further away from the trap axis, the RF trap frequency maybe cause large AC magnetic field. The author should give the shifts and uncertainty budget of magnetic field gradient and AC magnetic field.

As stated in the original manuscript, we suggest that the magnetic field averaged over all of the ion locations should be characterized with an uncertainty of $10\ \mu\text{G}$, and that the magnetic field gradient across the size of the ion crystal should be less than $1\ \text{mG}$, in order to avoid degrading the clock performance. The latter requirement corresponds to a fractional gradient smaller than $10^{-4}/\text{mm}$. We have added the following sentence to the manuscript: “Although it would be difficult to apply a $15\ \text{G}$ quantization field with such a $10^{-4}/\text{mm}$ fractional magnetic field gradient using standard Helmholtz coils, gradients as small as $10^{-6}/\text{mm}$ are routinely achieved using solenoids together with shim coils [13].”

We thank the reviewer for drawing our attention to the inhomogeneity of the AC magnetic field due to the ion trap. We have added the following sentence to the main text: “The ion trap should be designed to minimize the capacitance between electrodes in order to minimize the contribution to the second-order Zeeman shift due to the AC magnetic field at the trap drive frequency [14] (see Methods)”. We also added following paragraph to the Methods addressing this: “We estimate the size of this effect with the following simple model. We assume that the ion trap has a $500\ \mu\text{m}$ ion-to-electrode distance and is driven differentially with an RF voltage amplitude of $970\ \text{V}$, as is necessary to achieve the secular frequencies proposed here. Standard linear RF Paul traps constructed with cylindrical rods or blade electrodes that are electrically driven from one end have current running down the electrodes due which is sunk by a capacitance at the far ends of the electrodes of order $100\ \text{fF}$. At the center of the trap, symmetry dictates that the RF magnetic field should be zero. At the edge of the ion crystal, $22\ \mu\text{m}$ away from the trap center, this model predicts that there is an AC magnetic field of $50\ \text{mG}$, which causes an inhomogeneous frequency shift of $4\ \text{mHz}$. It should be possible to characterize the average over all ions of this shift sufficiently well as described in the text, and this is compatible with probe durations up to about $100\ \text{s}$. To achieve longer probe durations it will be necessary to design traps with less capacitance or more symmetric current distributions.”

5. Fig2(c) and line 266: When discussing the uncertainty of the probe laser Stark shift, the uncertainty is a few orders of magnitude smaller than the shift, how is this achieved? By measuring the shift by comparing the transition frequency between the increased power situation and the normal power situation? Or by hyper Ramsey scheme?

We anticipate using hyper-Ramsey or other protocols, and have modified the sentence to read: “We conservatively estimate that this can be characterized by an uncertainty of 2.0×10^{-20} by using hyper-Ramsey [15] or auto-balanced Ramsey [16] interrogation protocols.”

6. Line 221: As far as I know, the Doppler cooling efficiency is very sensitive to the power of the cooling laser, for such a large ion crystal, the laser power on each every ion should be different, thus the saturation parameter is not always 20, I don't know if the authors have already considered this in their calculations.

Although the Doppler temperature limit and cooling rate do depend somewhat on the laser power, this is a relatively minor effect, especially in the power broadened regime where we propose operating. For an elliptical Gaussian beam with waists of $100\ \mu\text{m}$ and $2\ \text{mm}$ propagating perpendicular to the trap axis, the laser intensity varies by $10\ \%$ across the size of the ion crystal. This intensity variation would cause the power broadened linewidth of the cooling transition to vary by $5\ \%$ across the ion crystal, with a similar variation in temperature limit. This is much smaller than the $30\ \%$ uncertainty we assume for the characterization of the secular motion temperature.

7. Line 275: I'm not sure if I have some misunderstanding on this: for single photon approach, the experiment requires a magnetic field of $15\ \text{G}$, and have a field inhomogeneity of $1\ \text{mG}$ over the size of $1\ \text{mm}$. It seems to me that this is very difficult to be achieved by simply adding a pair of Helmholtz coils.

See response to comment 4.

8. Line 428: As mentioned earlier, I think it is very difficult to be achieved for making a very strong magnetic field with a very small gradient like $1\ \text{mG}$ over the size of $1\ \text{mm}$, $10\ \text{micro-G}$ is even more difficult.

See response to comment 4. Note in particular that the field does not need to be uniform to $10\ \mu\text{G}$, but rather the average field over all of the ion locations needs to be characterized at this level.

9. Line 460: “assuming a conservative $1\ \text{s}$ probe duration and $80\ \%$ duty cycle.” Although the lifetime of $3\text{P}0$ clock state is quite long $106\ \text{s}$ at $15\ \text{G}$ magnetic field, ideally one can get very long coherence time, but actually the coherence time is determined by high performance of clock laser and ion heating during interrogation time. What is the evidence for $1\ \text{s}$, is it based on the experience of the Al^+ ion clock or have you done the calculation and analysis?

It seems to me that the probe time can be even longer.

While we have not done a rigorous calculation of the achievable laser coherence time, 1 s at the transition frequency of Sn^{2+} is well within the capabilities of state-of-the-art clock lasers [17], which have achieved significantly higher Q spectroscopy. Furthermore, new spectroscopy protocols such as correlation spectroscopy [18] and differential spectroscopy [19] enable probing beyond the laser coherence time, and may support probe times of many seconds with Sn^{2+} .

10. Line 472: “It may be possible. . . maintaining sub- 10^{-19} inaccuracy.” The author proposal using Sr^+ ion to co-trapping Sn^{2+} ions, the outside trapped Sr^+ can push the Sn^{2+} ions towards the trap axis and make the micromotion shift smaller. The issue is, it seems not have a “magic wavelength”, cooling Sr^+ ion during clock interrogation can lead into ac Stark shifts. Is the cooling laser off when probing? If so, the authors should consider the heating effect.

Yes, the Sr^+ cooling laser will cause a Stark shift on the Sn^{2+} clock transition if the cooling is applied simultaneously with the clock interrogation. For Sr^+ Doppler cooling on the 422 nm transition with a typical saturation parameter of 0.1, and using the value of the Sn^{2+} differential polarizability at 374 nm (which should be slightly larger than that at 422 nm), the fractional frequency shift of the Sn^{2+} clock transition is 5×10^{-19} . It should be possible to stabilize and characterize this shift significantly better, so we do not anticipate that this would be a problem. If it is, the cooling can be switched off during the clock interrogation.

III. REVIEWER # 2

dear editor,

the paper proposed by Leibrandt et al is a proposal for an optical atomic clock that could be operated with a cloud of one thousand ions as a base, whereas so far, trapped ion based optical clock are operated with a single ion. The idea is to make a more stable clock by interrogation of a thousand ions at the same time and then reduce the quantum projection noise which is the limiting factor for short term stability. The issue is to do that without degrading the accuracy and precision of the clock as the inhomogeneity of the ion environment can induce non uniform shifts responsible for a line broadening and an uncertainty in the shift estimation.

This idea is not new and several ions have been identified that share some characteristics that make them good candidates for this kind of set-up. The main figure of merit is the differential polarizability of the transition. If it is negative, two major effects cancel each-other, provided the RF-trap is operated at a proper frequency, which depends on this difference. This is covered by ref 17-21. The new results of this paper is to add an ion to this list : Sn^{2+} , which have some technical advantage over recent proposals, like the accessibility of the transition wavelength, but requires a strong magnetic field to mix different atomic fine-structure states and make the clock transition accessible to laser excitation. The best performances are reached by a compromise between the lowest possible magnetic field to control Zeeman shifts and the lowest possible laser power to control light induced shifts. If an experiment is able to follow these prescriptions without excess inhomogeneity, the expected clock performance are very good compared to the state of art in terms of uncertainty and inaccuracy.

We thank the referee for the report and agree completely that the novelty of the papers is proposing the Sn^{2+} clock and demonstrating that it has a number of advantages that may allow to achieve sub- 10^{-19} inaccuracy and a stability of the best optical lattice clocks, but in a system with much longer lifetime. To the best of our knowledge, this is also the first full-scale simulation of micromotion for a 1000-ion clock, which has required high-performance computing resources to perform a detailed modeling of the shifts under various experimental conditions. We will address all of the specific comments below and believe that following the referees’s advice has significantly improved the clarity of the paper.

We have broken up some of reviewer 2’s paragraphs and added numbers below so that they can be easily referenced.

1. At this stage of the paper, the reader still do not know what isotope is considered. We can guess that an even isotope is assumed but nothing is said about their abundance and possible choices. The calculations used for atomic structure calculations is now well established and one of the author is a reference in the optical clock community for the relevance of her methods.

We have modified the description of the magic trap drive frequency to make our choice of isotope clear: “All even isotopes of Sn^{2+} are equally suitable for clock operation, and different choices will have slightly different magic drive frequencies and laser cooling dynamics. The calculations throughout this article use $M_i \approx A m_p$ with $A = 118$ (where m_p

is the proton mass), resulting in $\Omega_m/(2\pi) = 225(5)$ MHz.”

2. The discussion regarding the clock operation in the context of a thousand trapped ion is far less convincing. There are a lot of statements regarding the cooling efficiency that look like speculations and are not supported by comparison with published and cited work. If the statements are based on molecular dynamics simulations, it is not written clearly and the fact that 3 laser beams with different directions are required is not justified anywhere. Indeed, as far as I know, no 3D Coulomb crystal experiments use this set-up as the micro-motion induced Doppler effect make Doppler cooling less efficient (see url = <https://link.aps.org/doi/10.1103/PhysRevA.105.023101> for the 2D case where different laser detunings are needed). A single one propagating along the axis is often the best solution for large 3D system.

As the reviewer points out, micromotion does make the Doppler cooling of large ion crystals less efficient for laser beams with nonzero components of their k-vector along the radial trap directions. However, for low-temperature ions in a perfectly harmonic trap, cooling with a single laser perfectly along the axial direction does not cool the two center-of-mass radial modes at all. In the presence of finite temperature and trap imperfections, cooling of the radial modes is still inefficient and the radial mode temperatures are higher than the Doppler limit, which would be problematic for minimizing the Doppler shift. Thus, here we propose using three orthogonal cooling beams. In this proposal, the high rf trapping frequency and narrow cooling transition linewidth help to make the radial cooling beams effective. This is both because the micromotion modulation index scales like the reciprocal of the trap frequency and because the micromotion sideband transitions are very well spectrally separated from each other and from the micromotion carrier transition (this is a distinct regime from that of PRA **105**, 023101 to which the reviewer refers).

For ions that are 22 μm away from the trap axis (i.e., the larger crystal radius), micromotion modulates the electric field of the purely radial vertical cooling beam with a modulation index of 2.4. The laser intensity for driving the micromotion carrier transition is reduced to 0.04 % of that for ions on the trap axis, and the laser intensity for driving the first (second) micromotion sideband transition is 26 % (19 %) of the carrier for ions on the trap axis. Cooling with a laser intensity of 0.04 % for the outermost ions will change the Doppler limit for the radial modes, but may still be ok because most of the ions will be cooled with a significantly higher intensity. If this turns out to be a problem, the cooling can be made more efficient by adding light that is red detuned from the first micromotion sideband to the radial beams. The latter technique has been successfully used in the NIST Ion Storage Group for single ion traps with large uncompensable micromotion.

As the reviewer suggests, a detailed molecular dynamics model of the laser cooling will be necessary to optimize the cooling parameters and to provide a rigorous constraint on the corresponding Doppler shift uncertainty of the clock, but this is beyond the scope of this manuscript and will be the subject of future work. However, it is unlikely that the detailed model will yield an ion temperature or Doppler shift uncertainty that is significantly larger than those of the simple model.

We have added the following sentence to the text: “A more complete molecular dynamics model incorporating the details of the laser cooling can be used to optimize the cooling parameters and is likely necessary to provide a rigorous constraint on the corresponding Doppler shift uncertainty of the clock.”

3. Other estimations lack an established support like the requirement for the magnetic field homogeneity. It is assumed to be better than 0.01% over a mm scale. It should be supported by results coming from similar experiments proving the feasibility of this requirement.

As stated in the original manuscript, we suggest that the magnetic field averaged over all of the ion locations should be characterized with an uncertainty of 10 μG , and that the magnetic field gradient across the size of the ion crystal should be less than 1 mG, in order to avoid degrading the clock performance. The latter requirement corresponds to a fractional gradient smaller than $10^{-4}/\text{mm}$. We have added the following sentence to the manuscript: “Although it would be difficult to apply a 15 G quantization field with such a $10^{-4}/\text{mm}$ fractional magnetic field gradient using standard Helmholtz coils, gradients as small as $10^{-6}/\text{mm}$ are routinely achieved using solenoids together with shim coils [13].” For more details about the magnetic field homogeneity, see the response to reviewer 1, comment 4.

4. A lot of discussion regarding the cloud shape and ion dynamics assume operating parameters that are justified a lot later without any notice. This is very detrimental. The RF optimum frequency is so large and the chosen single-ion secular frequencies are such that they correspond to very low Mathieu parameter, of the order of 10^{-2} . This is at the edge of the stability diagram for an RF-trap and to make it convincing, this proposal

should cite experimental results where such a low parameter has been operated with success. If these works do not exist yet, it is because it is really a challenge and this should not be considered as obvious in the article.

The trapping parameters proposed here correspond to Mathieu parameters of $|q_{x,y}| = 0.0127$, $a_x = 2.33 \times 10^{-5}$, and $a_x = -2.41 \times 10^{-5}$. While this is indeed a smaller q parameter than typical experiments, it is not closer to the edge of the stability region than typical experiments. For small q , the border of the stability diagram is given by $|a| < q^2/2$ [20]. Here, $|a|/(q^2/2) = 0.29 \ll 1$. For comparison, in Ref. [7] $|a|/(q^2/2) = 0.22$ and in Ref. [21] $|a|/(q^2/2) = 0.52$ are used experimentally without problems. Furthermore, our molecular dynamics calculations in the full-time-dependent potential do not indicate any instability issues. We have made this point more direct by modifying the last sentence of the paragraph describing our calculations to read “Second, we calculate the amplitude and direction of the micromotion of each ion and verify the stability of the ion crystal by time evolving the full equations of motion (without taking the pseudopotential approximation), starting from the ion positions computed in the first step.”

In fact, smaller q offers several advantages for Coulomb crystal clocks as the ions are deeper in the pseudopotential approximation. First, molecular dynamics simulations show that rf heating of large ion crystals, which arises due to the breakdown of the pseudopotential approximation $q \ll 1$, scales roughly like q^6 at low temperatures [11]. Extrapolating the results of Ref. [11] to $\Omega = 2\pi \times 225$ MHz, $q = 0.0127$, and an initial ion temperature of 120 μ K suggests that the rf heating might be of order 10^{-12} μ K/s, which is completely negligible even for probe durations of many seconds.

We have added the following to the manuscript: “These secular frequencies correspond to Mathieu parameters of $|q_{x,y}| = 0.0127$, $a_x = 2.33 \times 10^{-5}$, and $a_x = -2.41 \times 10^{-5}$, and can be achieved by differentially driving a trap with 500 μ m ion-to-electrode distance with 970 V amplitude of rf. This smaller-than-typical q parameter offers several advantages for Coulomb crystal clocks. Molecular dynamics simulations show that rf heating of large ion crystals, which arises due to the breakdown of the pseudopotential approximation $q \ll 1$, scales roughly like q^6 at low temperatures [11]. Care must be taken, however, to minimize anharmonicity of the trap potential as this can lead to additional rf heating [22]. Ion heating due to background gas collisions is also reduced for smaller q parameters [12].”

5. The part concerning the different spatial configuration of an ion ensemble is very confusing as the cited works all set the ions in a static pseudo-potential (I suggest the authors to read also Calvo and Yurtsever, EPJD (2007) DOI: 10.1140/epjd/e2007-00137-2), and the authors also use the same strategy, so why mentioning time-averaged kinetic energy of the micro-motion? This is the source of the pseudo-potential and it has nothing to do in a pseudo-potential description, it is one or the other, not both. How micro-motion is obtained in the pseudo-potential approximation? This point deserve clarification. The description of the forces taken into account should be more rigorous and corrected. Indeed, more details about the numerical simulations should be given (like in a methods part) to appreciate the relevance of the results and more earlier works should be cited to support the chosen strategy (see DOI: 10.1103/PhysRevA.82.033406 and <https://doi.org/10.1103/PhysRevLett.70.818> to start with). What about the choice of the damping constant simulating laser cooling? If it is too strong, the system is frozen in a non-equilibrium state. How the authors proceed to make sure this is not the case and that the motional temperature is 120 microK (cf caption of fig 4) ?

We thank the reviewer for making us aware of the interesting results in Ref. [23], and have added this to the manuscript. We have also added Refs. [24, 25] as examples of earlier molecular dynamics calculations following similar methods.

We have clarified the description of our molecular dynamics calculation method as follows: “First, similar to Refs. [26, 27], we calculate the equilibrium ion positions by time-evolving the equations of motion in the pseudopotential approximation starting from random initial positions and including a viscous damping force that qualitatively represents laser cooling. For each set of trap parameters and number of ions investigated, we generate up to 36 configurations in parallel using a high-performance computer cluster. For ion numbers up to 1000, this takes less than one week. The strength of the damping force is such that the cooling time constant is roughly 100 μ s, which is a typical experimental cooling time constant, and we have verified that reducing this strength by an order of magnitude does not reduce the number of distinct configurations. Once the equilibrium ion positions are known, it is straightforward to compute the frequencies and eigenvectors of the normal modes of secular motion and their Lamb-Dicke parameters [28, 29], and the spectrum of the clock transition [30]. Second, we calculate the amplitude and direction of the micromotion of each ion and verify the stability of the ion crystal by time evolving the full equations of motion (without taking the pseudopotential approximation), starting from the ion positions computed in the first step.”

We have also added a sentence about the cooling strength: “The strength of the damping force is such that the cooling time constant is roughly 100 μ s, which is a typical experimental cooling time constant. We have veri-

fied that reducing this strength by an order of magnitude does not reduce the number of distinct configurations.”

We hope that added text clarifies our approach. As we use well-established methods in these numerical simulations, we feel that an additional methods section would not add significant value.

6. The paragraph starting at line 393 is completely speculative as nothing proves that the listed requirements are compatible with each other. It is surprising to read that “maximize the lowest secular frequency” is in the list whereas a so small q_x parameter has been chosen. So this is certainly not the case. Line 413 mention a trade off to reach so small values, so the most important conditions should be enlightened and this need for a maximization should be justified.

Avoiding secular frequencies that are too low is important both because ion heating rates due to anomalous electric field noise are larger for smaller motional frequencies and because off-resonant excitation of low frequency motional sidebands during clock spectroscopy can lead to line pulling. The latter concern is briefly discussed in the previous paragraph: “The lowest frequency motional sideband transitions are above 14 kHz for any of the generated crystal configurations, and should not lead to significant line-pulling due to off-resonant excitation for probe times longer than about 10 ms.” We have added a new phrase to the manuscript about the former concern: “. . . low frequency motional modes should be avoided because they have higher heating rates due to electric field noise [31]”

However, as has been previously discussed, operating at low Mathieu q parameter is also advantageous, and furthermore it would be difficult to achieve significantly higher Mathieu q parameters with realistic rf voltage amplitudes.

Our intent with this paragraph is to motivate our choice of parameters, which are compatible with each other and enable both high accuracy and high stability clock performance. We have modified the final sentence of this paragraph accordingly to read “Single-ion secular frequencies of 1.15 MHz, 0.85 MHz, and 0.10 MHz represent a trade-off of the aforementioned optimization goals that enable both high accuracy and high stability clock performance.” We agree with the referee that a different set of trap parameters may be chosen, and could may lead to even higher performance. An exhaustive optimization of trap parameters is beyond the scope of this manuscript.

7. On the step of line 458, the requirements concerning the stability and the precision/accuracy of the proposed clock should be clearly identified and separated as some technical choices can be good for one figure of merit but bad for another.

We have added the sentence: “It may be possible to achieve a smaller secular motion and thus total systematic uncertainty by reducing the number of ions, but this would come at the cost of degraded stability.”

8. From line 472, another list of speculations appear, concerning the best side ion to be trapped with Sn^{2+} , to compress the ion cloud around the trap axis. The obvious solution would be to increase the cloud density, but once again, the reason why such a low Mathieu parameter is chosen is not given. Furthermore, this idea of using an outer cloud shell to compress the inner cloud is not obvious at all and is not supported by any calculations or previous work. This naive view seems in contradiction with the observation reported in PRL.86.1994.

We have addressed our choice of low Mathieu q parameter above.

We agree that the phrase “the Sr^+ ions would reside on the outside edge of the crystal, pushing the Sn^{2+} ions towards the trap axis” in the manuscript might have been misleading. We have changed this language to “the Sr^+ ions experience a weaker radial restoring force from the trap and thus fill the Coulomb crystal lattice sites further from the trap axis, causing the Sn^{2+} ions to reside in a cylindrical volume of lattice sites closer to the trap axis [32]”. In Fig. 2 of Ref. [32] that the referee refers to above, it is clearly shown that the lower mass-to-charge ratio ion ($^{24}\text{Mg}^+$) occupies a cylindrical volume of Coulomb crystal lattice sites near the trap axis, while the higher mass-to-charge ratio ion ($^{40}\text{Ca}^+$) resides in the lattice sites further from the trap axis to fill out the ellipsoidal two-species crystal shape.

9. As for the idea of using a multipole trap, I’d like to point to the authors that ref 54 and 56 concern room temperature large ion clouds and in the cold limit, the self organization of ions is very different (see for example J. Phys. B: At. Mol. Opt. Phys. 42 (2009) 154002). For smaller sample, the use of multipole trap is described here : DOI: 10.1103/PhysRevA.81.043410 . Contrary to what is stated on line 489-494, ions will not gather closer to the axis, because of Coulomb repulsion. (read again section V of ref 17) Anyway, and unfortunately, using a multipole trap with laser cooled ions may be a wrong good idea as stated in this series of paper :

<https://doi.org/10.1080/09500340.2017.1408866> and M Marchenay et al 2021 Quantum Sci. Technol. 6 024016 .

We thank the reviewer for pointing out these references. Based on the calculations in Ref. [33], the second-order Doppler shift due to micromotion averaged over all of the ions is still smaller in a multipole trap than in a quadrupole trap, but it is true that the nonuniform radial density distribution does reduce the benefit. As the reviewer points out, more calculations are needed to determine if using multipole traps is truly beneficial for these systems, so we have removed this suggestion from the manuscript.

10. At this stage, I think the reader is kind of lost with the come back of spectroscopic considerations from line 495. This would be better placed at the beginning of the paper or it could be suppressed as nothing new is brought here.

We have carefully considered the placement of the section “Sensitivity to beyond Standard Model physics”. The introduction primarily provides context in the form of an overview of various clock platforms and introduces the potential benefits and challenges of Coulomb crystal clocks, such as the issue of the micromotion. Adding a section about clock applications here would be somewhat confusing and would make the introduction section too long. Placing it between “Sn²⁺ atomic properties” and “Sn²⁺ clock operation” will be confusing as it will break up the discussion of Sn²⁺ properties that are critical to clock operation and the assessment of systematics and the discussion of clock operation. Placing it later will break up the discussion of Sn²⁺ clock operation and resulting clock performance. As this section deals with applications of the Sn²⁺ clock, the present placement after the discussion of clock operation and performance and before the conclusion appears to be most suitable. The sentence at the end of the introduction “We conclude by discussing prospects for fifth force searches using isotope shift measurements, as well as other tests of fundamental physics with Sn²⁺.” alerts the reader that this discussion is placed at the end of the paper.

We also believe that it is important to keep this discussion in the paper due to the present status of the fifth force searches with the non-linearity of King plots. Yb results already show a non-linearity which is most likely from the nuclear deformation. However, the data indicate that non-linearity may come from at least two sources. Ca data do not show non-linearity at present. Therefore, it is important to find another suitable system lighter than Yb where such experiments may be performed. Removing dependence on the nuclear deformation requires the use of four or more even isotopes, and Sn²⁺ is unique in having so many suitable isotopes.

As a conclusion, I think this paper is not ready yet for publication. Furthermore, I also think that the presented results are not noteworthy neither original enough for publication by nature communication.

We have addressed in detail all of the referee’s comments. The thoughtful and constructive comments of all three referees helped us improve the paper, providing more clarity on the choice of the specific clock operating parameters as well as assessing other potential systematic effects. Solving the question of significant improvement of trapped ion clock statistics while keeping systematic uncertainties under 10^{-19} is of critical importance to metrology. Our work provides a pathway towards a solution of this problem with a new system, Sn²⁺, and provides a detailed investigation of micromotion suppression with 1000 ions using high-performance computer simulations. The Sn²⁺ ion system also provide a way to resolve a present conundrum with fifth force searches with isotope shift measurements. Therefore, we believe the paper satisfies the Nature Communications standards of novelty and importance.

IV. REVIEWER # 3

The manuscript “Prospects of a thousand-ion Sn²⁺ Coulomb-crystal clock with sub- 10^{-19} inaccuracy”, gives a detailed account on the possibility of building a highly accurate atomic clock based on a large three dimensional ion Coulomb crystal. By considering a clock based on up to 1000 ions, the authors’ goal is to produce an atomic clock based on ions, which not only can compete with neutral atom lattice clocks with respect to total achievable inaccuracy, but also with respect to needed experimental integration time. The authors have made a great effort to assess many of the potential limiting issues related to the specific ion and trapping configuration in mind. The manuscript is organized in a logical way by first describing the reason to consider the Sn²⁺ ion species and the expected properties of the relevant electronic levels, followed by presenting how the trapping fields and properties of and ion motion within a Coulomb crystal influence the accuracy of the Coulomb crystal clock. Finally, before a summary and outlook, a discussion on various limiting factors and potential further improvements are briefly presented.

All in all, after a revision of the manuscript with respect to the comments and suggestions stated below, I find the

manuscript suitable for publication in Nature Communication, since the work represents a promising scenario for a future new improved time standard.

We thank the reviewer for supporting publication of our manuscript in Nature Communications.

Comments/questions:

1) In Fig. 1, a Coulomb crystal of 1000 ions is presented. It is obvious that a very elongated crystal along the rf field-free axis is preferable, but it is less clear why trap parameters giving rise to a relatively large difference in the crystal dimensions in the perpendicular plane is chosen. Perhaps the authors could elaborate a bit on that.

In the limit of equal trapping strength in the two radial dimensions, which would lead to equal crystal dimensions in the two radial dimensions, there is a zero frequency motional mode in which the crystal rotates about the trap axis. This mode would be difficult to cool. We have added the following to the manuscript to address this: “The two radial single-ion secular frequencies must be different in order to avoid a zero-frequency mode in which the ion crystal rotates about the trap axis.”.

2) In Fig. 1, the colors of the clock and cooling beams are very close to each other. Choosing a different color for the cooling beams would be helpful also for the readability of the text on the illustration.

We thank the reviewer for pointing this out and have made the colors more distinct.

3) Since the “magic” rf frequency of 225 MHz is rather high, a reference that shows this should be achievable with the trap dimensions in mind would be needed.

We have added Ref. [34], in which an ion trap with 160 μm ion-to-electrode distance is driven at 240 MHz with up to 1 kV amplitude of rf. The manuscript now reads: “Although this is a high trap frequency, previous experiments have operated in this range [34].”

4) In fig. 2.c, the lines indicating the uncertainties of the second order Zeeman shifts and Stark uncertainties translates into a magnetic and laser field instabilities of 10^{-5} and 10^{-4} , respectively. It would be needed with some references that such instabilities should be achievable over the total volume of the Coulomb crystal.

In the original manuscript, we suggest that the magnetic field averaged over all of the ion locations should be characterized with an uncertainty of 10 μG , and that the magnetic field gradient across the size of the ion crystal should be less than 1 mG, in order to avoid degrading the clock performance. The latter requirement corresponds to a fractional gradient smaller than $10^{-4}/\text{mm}$. We have added the following sentence to the manuscript: “Although it would be difficult to apply a 15 G quantization field with such a $10^{-4}/\text{mm}$ fractional magnetic field gradient using standard Helmholtz coils, gradients as small as $10^{-6}/\text{mm}$ are routinely achieved using solenoids together with shim coils [13].”

Reviewer 1 drew our attention to the inhomogeneity of the AC magnetic field due to the ion trap. We have added the following sentence to the main text: “The ion trap should be designed to minimize the capacitance between electrodes in order to minimize the contribution to the second-order Zeeman shift due to the ac magnetic field at the trap drive frequency [14] (see Methods)”. We also added following paragraph to the Methods addressing this: “We estimate the size of this effect with the following simple model. We assume that the ion trap has a 500 μm ion-to-electrode distance and is driven differentially with an RF voltage amplitude of 970 V, as is necessary to achieve the secular frequencies proposed here. Standard linear RF Paul traps constructed with cylindrical rods or blade electrodes that are electrically driven from one end have current running down the electrodes due which is sunk by a capacitance at the far ends of the electrodes of order 100 fF. At the center of the trap, symmetry dictates that the RF magnetic field should be zero. At the edge of the ion crystal, 22 μm away from the trap center, this model predicts that there is an AC magnetic field of 50 mG, which causes an inhomogeneous frequency shift of 4 mHz. It should be possible to characterize the average over all ions of this shift sufficiently well as described in the text, and this is compatible with probe durations up to about 100 s. To achieve longer probe durations it will be necessary to design traps with less capacitance or more symmetric current distributions.”

The hyper-Ramsey and auto-balanced Ramsey techniques relax the requirement for laser intensity stability and we anticipate using hyper-Ramsey or other protocols. We have modified the sentence to read: “We conservatively estimate that this can be characterized by an uncertainty of 2.0×10^{-20} by using hyper-Ramsey [15] or auto-balanced Ramsey [16] interrogation protocols.”

5) With respect to the assessment of time dilation and rf field induced Stark shifts, the authors have considered 36 stationary configurations of the Coulomb crystals which seems reasonable. A single background gas collisions may though lead to dynamical changes in the crystal structure or even lead to crystal melting. Only very few experimental investigations have been made so far to address these issues quantitatively, but they could potentially constitute limitations to use larger crystals in particularly in room temperature trapping environments. Please comment on this in a revised manuscript.

We agree that this experiment most likely needs to use a cryogenic ion trap in which the background gas pressure and thus collision rate can be greatly reduced. We have added the following paragraph to the methods: “One can estimate the background gas collision shift based on scaling analysis of the shift for single-ion clocks [5]. The total rate of Langevin collisions between a Coulomb crystal and background gas molecules scales linearly with the number of ions and quadratically with the charge of the ions, so for a 1000 ion Sn^{2+} clock the background gas collision rate will be approximately 4000 times larger than in single-ion clocks. As a conservative upper bound, we take that the systematic uncertainty due to background gas collisions scales linearly with the collision rate. A more detailed model would take into account that only the Doppler shift component of the uncertainty scales with the total collision rate while the phase shift component of the uncertainty scales with the collision rate per ion. Current state-of-the-art single-ion clocks operating in room temperature ultrahigh vacuum (UHV) chambers with background gas pressures of roughly 10^{-10} torr have systematic uncertainties due to background gas collisions of order 10^{-19} [6, 7]. Although background gas collisions would likely contribute significant uncertainty for 1000 ion Sn^{2+} clocks in room temperature UHV chambers, pressures below 10^{-16} torr have been achieved in cryogenic vacuum chambers [8]. Thus, by operating Sn^{2+} clocks in cryogenic vacuum chambers it should be possible to suppress the collision shift uncertainty to below 10^{-21} .”

As ions are lost from the trap or if they undergo chemical reactions to become dark ions, the crystal configuration will change. However, one important result of our simulations is that the systematic shifts do not change at a level that would increase the overall uncertainty of the clock even as the ion crystal configuration changes. We have added the following sentence to the text to highlight this feature: “Importantly, the differences of these shifts for different ion crystal configurations are smaller than our estimated uncertainties, so Sn^{2+} clocks will be robust to ion loss and background gas collisions that cause the crystal to melt and recrystallize in a different configuration.”

6) With respect to computing the frequencies and eigenvectors of the normal modes of the secular motion and Lamb-Dicke parameters reference is given to [41] in the manuscript. Since this paper deals only with linear chains of ions, a more suitable reference for the current proposal could, e.g., be New J. Phys. 14 093023 (2012).

We thank the reviewer for making us aware of Ref. [29] and have added a citation in the manuscript.

7) Fig. 3, which is central to the manuscript, contains a lot of information presented in a condensation fashion. For the non-expert it can be very hard to be sure to have understood it fully as it is.

In fig. 3.b, it would be great to have a clearer physical description of what is meant by “net secular motion shift”. As it looks like it is independent of the number of ions, I assume it accounts only for the time dilation related to the secular motion? Could there also be a secular motion dependent Stark shift which should be accounted for?

We thank the reviewer for pointing out that we had neglected to provide a definition of the net secular motion shift. We have modified the caption of Fig. 3b to include “The net secular motion (micromotion) shift is defined as the sum of the negative time-dilation shift and the positive Stark shift caused by the electric field of the ion trap in the reference frame of the ion due to ion motion at the secular motional frequencies (integer multiples of the trap drive frequency).” Furthermore, we have modified the corresponding sentence in the main text to read “The net secular motion shift (defined as the sum of the time dilation shift and Stark shift due to ion motion at the secular frequencies) for 1000 ions is -2.4×10^{-19} , and we conservatively estimate that this can be characterized with an uncertainty of 7.2×10^{-20} .”

Fig. 3.d is very challenging. If I am right that the green bars are just the difference between the red and blue, I believe one could communicate that clearer. The two first panels deal with the secular motion, while the rest of the panels deal with rf induced micromotion. For the sake of clarity, it would be good to separate Fig. 3.d. into three separate sub-figures (d.,e., and f.) and not for all panels use the term “net micromotion shift”. In the last four panels the fractional frequency shifts exhibit large increases at the magical rf frequency. Why is this the case? This should be explained more carefully in the text. The assumed temperature for the secular contributions should be stated in the figure caption.

We thank the referee for pointing out that “net micromotion shift” is a misleading label for the green color in Fig. 3d into three subpanels. Although we have elected not to separate Fig. 3d, we have changed the label for the green color to read “net motional shift”, referring to either the net micromotion shift or the net secular motion shift.

The time dilation shift due to motion at the trap drive frequency is indeed much larger than that due to motion at other frequencies. This is because for laser cooled three-dimensional ion crystals the micromotion is much larger than the secular motion. Indeed, one of the contributions of this manuscript is to carefully calculate how well this very large time dilation shift due to micromotion can be cancelled by the corresponding trap Stark shift.

We have added a sentence to the Fig. 3b caption “The secular motion temperature here and in panel d is assumed to be 120 μK .”

8) Somewhere in the manuscript it would be great to have a short discussion of what would the influence of a radial temperature gradient in the crystal. A priori one cannot expect to be able to cool all ions in the crystal to the same final temperature.

We certainly agree that experimental imperfections will result in somewhat unequal temperatures of the individual motional modes. The model of cooling presented in the manuscript is intended to serve as an estimate of the magnitude of the secular motion Doppler shift averaged over all of the ions, and the achievable uncertainty in this shift, rather than to provide a rigorous evaluation of the secular motion Doppler shift uncertainty for a particular experimental implementation. This simple model indicates that the secular motion Doppler shift will not be a dominant limitation to clock uncertainty.

A detailed molecular dynamics model of the laser cooling will be necessary to optimize the cooling parameters and to provide a rigorous constraint on the corresponding Doppler shift uncertainty of the clock, but this is beyond the scope of this manuscript and will be the subject of future work. However, it is unlikely that the detailed model will yield an ion temperature or Doppler shift uncertainty that is significantly larger than those of the simple model.

We have added the following sentence to the text: “A more complete molecular dynamics model incorporating the details of the laser cooling can be used to optimize the cooling parameters and is likely necessary to provide a rigorous constraint on the corresponding Doppler shift uncertainty of the clock.”

For more discussion of the laser cooling, see also the response to reviewer 2 comment 2.

9) With respect to uncertainty in the shifts due to BBR, is an effective temperature with an uncertainty of 500 mK reasonable over the whole crystal lengths (1 mm)?

Well designed ion traps (with high electrical and thermal conductivity and low dielectric loss) have achieved smaller than 1 K thermal gradients across the entire trap [35, 36]. Even if this is not the case, it would be possible to measure the BBR temperature as a function of position within the trap by e.g. trapping a single Sr^+ ion and measuring the BBR Stark shift of its clock transition (which is much larger than Sn^{2+} and very well known [37]) for different positions within the trap. Finally, most likely it will be beneficial to operate Sn^{2+} clocks in cryogenic ion traps where the BBR shift will be negligible.

10) With respect to the two species Coulomb crystal discussion on p. 8 and 9, it should be noted that volume of the Sn^{2+} ions in this case will be a nearly perfect cylinder, and hence for a given crystal length and number of ions the outermost ions will be positioned closer to the trap axis when compared to the case of a single Sn^{2+} species ellipsoidal crystal. Consequently, this would lead to a smaller average time dilation and Stark shifts. A potential reference in this relation could, e.g., be Phys. Rev. Lett. 86, 1994 (2001).

We thank the referee for their suggestion, and have added the suggested reference [32] and a statement that the Sn^{2+} forms a cylindrical crystal along the trap axis.

[1] N.-H. Rehbehn, M. K. Rosner, H. Bekker, J. C. Berengut, P. O. Schmidt, S. A. King, P. Micke, M. F. Gu, R. Müller, A. Surzhykov, and J. R. C. López-Urrutia, Sensitivity to new physics of isotope-shift studies using the coronal lines of

- highly charged calcium ions, *Phys. Rev. A* **103**, L040801 (2021).
- [2] S.-Y. Liang, T.-X. Zhang, H. Guan, Q.-F. Lu, J. Xiao, S.-L. Chen, Y. Huang, Y.-H. Zhang, C.-B. Li, Y.-M. Zou, J.-G. Li, Z.-C. Yan, A. Derevianko, M.-S. Zhan, T.-Y. Shi, and K.-L. Gao, Probing multiple electric-dipole-forbidden optical transitions in highly charged nickel ions, *Phys. Rev. A* **103**, 022804 (2021).
 - [3] S. A. King, L. J. Spieß, P. Micke, A. Wilzewski, T. Leopold, E. Benkler, R. Lange, N. Huntemann, A. Surzhykov, V. A. Yerokhin, J. R. C. López-Urrutia, and P. O. Schmidt, An optical atomic clock based on a highly charged ion (2022).
 - [4] N. Kimura, Priti, Y. Kono, P. Pipatpakorn, K. Soutome, N. Numadate, S. Kuma, T. Azuma, and N. Nakamura, Hyperfine-structure-resolved laser spectroscopy of many-electron highly charged ions (2023).
 - [5] A. M. Hankin, E. R. Clements, Y. Huang, S. M. Brewer, J.-S. Chen, C. W. Chou, D. B. Hume, and D. R. Leibbrandt, Systematic uncertainty due to background-gas collisions in trapped-ion optical clocks, *Phys. Rev. A* **100**, 033419 (2019).
 - [6] N. Huntemann, C. Sanner, B. Lipphardt, C. Tamm, and E. Peik, Single-ion atomic clock with 3×10^{-18} systematic uncertainty, *Phys. Rev. Lett.* **116**, 063001 (2016).
 - [7] S. Brewer, J.-S. Chen, A. Hankin, E. Clements, C. Chou, D. Wineland, D. Hume, and D. Leibbrandt, $^{27}\text{Al}^+$ quantum-logic clock with a systematic uncertainty below 10^{-18} , *Phys. Rev. Lett.* **123**, 033201 (2019).
 - [8] G. Gabrielse, X. Fei, L. A. Orozco, R. L. Tjoelker, J. Haas, H. Kalinowsky, T. A. Trainor, and W. Kells, Thousandfold improvement in the measured antiproton mass, *Phys. Rev. Lett.* **65**, 1317 (1990).
 - [9] B. moo Ann, F. Schmid, J. Krause, T. W. Hänsch, T. Udem, and A. Ozawa, Motional resonances of three-dimensional dual-species coulomb crystals, *J. Phys. B: At. Mol. Opt. Phys.* **52**, 035002 (2019).
 - [10] M. W. van Mourik, P. Hrmo, L. Gerster, B. Wilhelm, R. Blatt, P. Schindler, and T. Monz, rf-induced heating dynamics of noncrystallized trapped ions, *Phys. Rev. A* **105**, 033101 (2022).
 - [11] V. L. Ryjkov, X. Zhao, and H. A. Schuessler, Simulations of the rf heating rates in a linear quadrupole ion trap, *Phys. Rev. A* **71**, 033414 (2005).
 - [12] C. B. Zhang, D. Offenber, B. Roth, M. A. Wilson, and S. Schiller, Molecular-dynamics simulations of cold single-species and multispecies ion ensembles in a linear paul trap, *Phys. Rev. A* **76**, 012719 (2007).
 - [13] J. W. Britton, J. G. Bohnet, B. C. Sawyer, H. Uys, M. J. Biercuk, and J. J. Bollinger, Vibration-induced field fluctuations in a superconducting magnet, *Phys. Rev. A* **93**, 062511 (2016).
 - [14] H. C. J. Gan, G. Maslennikov, K.-W. Tseng, T. R. Tan, R. Kaewuam, K. J. Arnold, D. Matsukevich, and M. D. Barrett, Oscillating-magnetic-field effects in high-precision metrology, *Phys. Rev. A* **98**, 032514 (2018).
 - [15] N. Huntemann, B. Lipphardt, M. Okhapkin, C. Tamm, E. Peik, A. V. Taichenachev, and V. I. Yudin, Generalized Ramsey excitation scheme with suppressed light shift, *Phys. Rev. Lett.* **109**, 213002 (2012).
 - [16] C. Sanner, N. Huntemann, R. Lange, C. Tamm, and E. Peik, Autobalanced Ramsey spectroscopy, *Phys. Rev. Lett.* **120**, 053602 (2018).
 - [17] D. Matei, T. Legero, S. Häfner, C. Grebing, R. Weyrich, W. Zhang, L. Sonderhouse, J. Robinson, J. Ye, F. Riehle, and U. Sterr, $1.5 \mu\text{m}$ lasers with sub-10 mhz linewidth, *Phys. Rev. Lett.* **118**, 263202 (2017).
 - [18] E. R. Clements, M. E. Kim, K. Cui, A. M. Hankin, S. M. Brewer, J. Valencia, J.-S. Chen, C.-W. Chou, D. R. Leibbrandt, and D. B. Hume, Lifetime-limited interrogation of two independent $^{27}\text{Al}^+$ clocks using correlation spectroscopy, *Phys. Rev. Lett.* **125**, 243602 (2020).
 - [19] M. E. Kim, W. F. McGrew, N. V. Nardelli, E. R. Clements, Y. S. Hassan, X. Zhang, J. Valencia, H. Leopardi, D. B. Hume, T. M. Fortier, A. D. Ludlow, and D. R. Leibbrandt, Optical coherence between atomic species at the second scale: improved clock comparisons via differential spectroscopy, arXiv:2109.09540 (2021).
 - [20] P. K. Ghosh, *Ion traps* (Clarendon Press, 1996).
 - [21] C. wen Chou, C. Kurz, D. B. Hume, P. N. Plessow, D. R. Leibbrandt, and D. Leibfried, Preparation and coherent manipulation of pure quantum states of a single molecular ion, *Nature* **545**, 203 (2017).
 - [22] J. Pedregosa, C. Champenois, M. Houssin, and M. Knoop, Anharmonic contributions in real rf linear quadrupole traps, *Int. J. Mass Spectrometry* **290**, 100 (2010).
 - [23] F. Calvo and E. Yurtsever, Non-monotonic size effects on the structure and thermodynamics of coulomb clusters in three-dimensional harmonic traps, *Eur. Phys. J. D* **44**, 81 (2007).
 - [24] J. P. Schiffer, Phase transitions in anisotropically confined ionic crystals, *Phys. Rev. Lett.* **70**, 818 (1993).
 - [25] M. Marcianti, C. Champenois, A. Calisti, J. Pedregosa-Gutierrez, and M. Knoop, Ion dynamics in a linear radio-frequency trap with a single cooling laser, *Phys. Rev. A* **82**, 033406 (2010).
 - [26] K. Arnold, E. Hajiyev, E. Paez, C. H. Lee, and M. D. Barrett, Prospects for atomic clocks based on large ion crystals, *Phys. Rev. A* **92**, 032108 (2015).
 - [27] G. A. Kazakov, J. Bohnet, and T. Schumm, Prospects for a bad-cavity laser using a large ion crystal, *Phys. Rev. A* **96**, 023412 (2017).
 - [28] D. F. V. James, Quantum dynamics of cold trapped ions with application to quantum computation, *Appl. Phys. B* **66**, 181 (1998).
 - [29] H. Landa, M. Drewsen, B. Reznik, and A. Retzker, Modes of oscillation in radiofrequency paul traps, *New J. Phys.* **14**, 093023 (2012).
 - [30] D. J. Wineland, C. Monroe, W. M. Itano, D. Leibfried, B. E. King, and D. M. Meekhof, Experimental issues in coherent quantum-state manipulation of trapped atomic ions, *J. Res. Natl. Inst. Stand. Technol.* **103**, 259 (1998).
 - [31] M. Brownnutt, M. Kumph, P. Rabl, and R. Blatt, Ion-trap measurements of electric-field noise near surfaces, *Rev. Mod. Phys.* **87**, 1419 (2015).
 - [32] L. Hornekær, N. Kjærgaard, A. M. Thommesen, and M. Drewsen, Structural properties of two-component coulomb crystals in linear paul traps, *Phys. Rev. Lett.* **86**, 1994 (2001).

- [33] C. Champenois, About the dynamics and thermodynamics of trapped ions, *J. Phys. B: At. Mol. Opt. Phys.* **42**, 154002 (2009).
- [34] S. R. Jefferts, C. Monroe, E. W. Bell, and D. J. Wineland, Coaxial-resonator-driven rf (paul) trap for strong confinement, *Phys. Rev. A* **51**, 3112 (1995).
- [35] M. Doležal, P. Balling, P. B. R. Nisbet-Jones, S. A. King, J. M. Jones, H. A. Klein, P. Gill, T. Lindvall, A. E. Wallin, M. Merimaa, C. Tamm, C. Sanner, N. Huntemann, N. Scharnhorst, I. D. Leroux, P. O. Schmidt, T. Burgermeister, T. E. Mehlstäubler, and E. Peik, Analysis of thermal radiation in ion traps for optical frequency standards, *Metrologia* **52**, 842 (2015).
- [36] T. Nordmann, A. Didier, M. Doležal, P. Balling, T. Burgermeister, and T. E. Mehlstäubler, Sub-kelvin temperature management in ion traps for optical clocks, *Rev. Sci. Instrum.* **91**, 111301 (2020).
- [37] P. Dubé, A. A. Madej, M. Tibbo, and J. E. Bernard, High-accuracy measurement of the differential scalar polarizability of a $^{88}\text{Sr}^+$ clock using the time-dilation effect, *Phys. Rev. Lett.* **112**, 173002 (2014).

Reviewer #1 (Remarks to the Author):

Now, I have read the answers and revised manuscript, the authors replied to all my questions and concerns. In my opinion, this new ion is a promising candidate for clock with sub-E-19 inaccuracy, the manuscript is well-written and suitable for publication in NC.

My only concern now is the shift caused by ions loss rate: as the author said "the differences of these shifts for different ion crystal configurations are smaller than our estimated uncertainties", I believe that, but what I am still concerned about is how much shift is caused by ion loss during a single probe cycle, and thus the asymmetry of the clock line shape, and whether it will be on the order of E-19.

Reviewer #2 (Remarks to the Author):

dear editor,

the authors have addressed all the questions raised by the reviewers and are documenting their assumptions with a lot of relevant references. I think this paper can now be published by nature communication, with a main interest of proposing high precision frequency measurements based on a 3D ion cloud. Nevertheless, I still have few comments for the authors:

the authors state that:

"To the best of our knowledge, this is also the first full-scale simulation of micro-motion for a 1000-ion clock, which has required high-performance computing resources to perform a detailed modeling of the shifts under various experimental conditions. "

Even if the purpose of the simulations were not oriented toward frequency metrology, a full-scale simulation of micromotion for a 1024-ion cloud can be found in

<https://journals.aps.org/pr/abstract/10.1103/PhysRevA.108.013109>

and

<https://doi.org/10.1063/5.0046693>

these last results were obtained with a single laser beam propagating along the trap axis. Regarding the need for different laser beams to cool a 3D sample, I refer the authors to the work done in the group of M. Drewsen where most of the time a single direction of propagation is used to cool a 3D cloud and it is along the trap axis when the cloud is 3D. In our group, we have tried to cool a large 3D cloud with a single beam having projection along the axial and radial trap axis and it was a fiasco. Everything went smooth when we moved the beam along the trap axis. So I think this laser beam propagation direction is still an open question and it is a very good thing if the authors try different spatial configurations in their set-up. They will bring a very useful knowledge to the community interested in using large 3D ion cloud.

Caroline Champenois, CNRS-AMU, France

Response to referee comments on manuscript “Prospects of a thousand-ion Sn^{2+} Coulomb-crystal clock with sub- 10^{-19} inaccuracy”

David R. Leibrandt,^{1,2,3,*} Sergey G. Porsev,⁴ Charles Cheung,⁴ and Marianna S. Safronova⁴

¹*Department of Physics and Astronomy, University of California, Los Angeles, California, 90095, USA*

²*Time and Frequency Division, National Institute of Standards and Technology, Boulder, Colorado, 80305, USA*

³*Department of Physics, University of Colorado, Boulder, Colorado, 80309, USA*

⁴*Department of Physics and Astronomy, University of Delaware, Newark, Delaware 19716, USA*

(Dated: December 1, 2023)

I. INTRODUCTION

We thank the referees for supporting publication of our manuscript in Nature Communications, and for their comments that have helped us to further improve the manuscript. In the following, we reply to the referees’ comments point by point. The referee comments are reproduced verbatim in gray text, followed by our responses in black text. We have appended a copy of the revised manuscript with the changes highlighted.

II. REVIEWER # 1

Now, I have read the answers and revised manuscript, the authors replied to all my questions and concerns. In my opinion, this new ion is a promising candidate for clock with sub-E-19 inaccuracy, the manuscript is well-written and suitable for publication in NC.

We thank the reviewer for their positive assessment of our work.

My only concern now is the shift caused by ions loss rate: as the author said “the differences of these shifts for different ion crystal configurations are smaller than our estimated uncertainties”, I believe that, but what I am still concerned about is how much shift is caused by ion loss during a single probe cycle, and thus the asymmetry of the clock line shape, and whether it will be on the order of E-19.

We thank the reviewer for calling attention to this important point. Previously, this point was only addressed briefly in the “Background gas collision shift” section of the methods without being called out in the main text. We have now added the sentence “It may, in fact, be necessary to use a cryogenic vacuum chamber to suppress the uncertainty due to background gas collisions to a negligible level (see Methods).” to the main text (line 514) and expanded the “Background gas collision shift” section of the methods to read as follows.

“A detailed analysis of the background gas collision shift and uncertainty for many ion clocks requires computationally intensive Monte Carlo simulations and is beyond the scope of this article, but in the following we provide a simple estimate based on scaling an analysis of the uncertainty for single-ion clocks [1]. The total rate of Langevin collisions between a Coulomb crystal and background gas molecules scales linearly with the number of ions and linearly with the charge of the ions, so for a 1000 ion Sn^{2+} clock the background gas collision rate will be approximately 2000 times larger than in single-ion clocks. As a conservative upper bound, we suppose that the systematic uncertainty due to background gas collisions scales linearly with the collision rate. A more detailed model would take into account that only the Doppler shift component of the uncertainty scales with the total collision rate while the phase shift component of the uncertainty (which dominated the collision shift uncertainty in Ref. [1]) scales with the collision rate per ion. Current state-of-the-art single-ion clocks operating in room temperature ultrahigh vacuum (UHV) chambers with background gas pressures of roughly 10^{-10} torr have systematic uncertainties due to background gas collisions of order 10^{-19} [2, 3]. Although according to this conservative estimate background gas collisions may contribute of order 10^{-18} uncertainty for 1000 ion Sn^{2+} clocks in room temperature UHV chambers with pressures of at best 10^{-12} torr, Monte Carlo simulations will likely be able to constrain the uncertainty to be much smaller.

* leibrandt@physics.ucla.edu

Alternatively, pressures below 10^{-16} torr have been achieved for ion traps operated in cryogenic vacuum chambers [4]. An even more conservative upper bound can be obtained by considering the rate of collisions as follows. For 10^{-16} torr of He background gas at 10 K, the total rate of Langevin collisions with any of the ions in a 1000 Sn^{2+} ion crystal is $1 \times 10^{-4}/\text{s}$. For a 1 s probe time, this amounts to one collision every 10^4 probes. At most, this could result in a fractional shift of the measured transition frequency of 10^{-4} relative to the spectroscopic linewidth, which in this case is 1 Hz. Thus, the background gas collision shift can be straightforwardly upper bounded to be below 10^{-19} for Sn^{2+} clocks operating in cryogenic vacuum chambers. In practice, collisions that deposit a significant amount of kinetic energy to the ion crystal will not contribute to the clock spectroscopy signal due to the Debye-Waller reduction of the Rabi frequency [1], so the actual shift and its uncertainty will be much smaller.”

III. REVIEWER # 2

dear editor,

the authors have addressed all the questions raised by the reviewers and are documenting their assumptions with a lot of relevant references. I think this paper can now be published by nature communication, with a main interest of proposing high precision frequency measurements based on a 3D ion cloud. Nevertheless, I still have few comments for the authors:

We thank the reviewer for supporting publication of our manuscript in Nature Communications.

the authors state that:

“To the best of our knowledge, this is also the first full-scale simulation of micro-motion for a 1000-ion clock, which has required high-performance computing resources to perform a detailed modeling of the shifts under various experimental conditions.”

Even if the purpose of the simulations were not oriented toward frequency metrology, a full-scale simulation of micromotion for a 1024-ion cloud can be found in

<https://journals.aps.org/pr/abstract/10.1103/PhysRevA.108.013109>

and

<https://doi.org/10.1063/5.0046693>

these last results were obtained with a single laser beam propagating along the trap axis. Regarding the need for different laser beams to cool a 3D sample, I refer the authors to the work done in the group of M. Drewsen where most of the time a single direction of propagation is used to cool a 3D cloud and it is along the trap axis when the cloud is 3D. In our group, we have tried to cool a large 3D cloud with a single beam having projection along the axial and radial trap axis and it was a fiasco. Everything went smooth when we moved the beam along the trap axis. So I think this laser beam propagation direction is still an open question and it is a very good thing if the authors try different spatial configurations in their set-up. They will bring a very useful knowledge to the community interested in using large 3D ion cloud.

Caroline Champenois, CNRS-AMU, France

Please accept our apologies for not being aware of the simulations of large ion crystals including micromotion in Refs. [5, 6]. We have added citations to these papers in this sentence of the revised manuscript: “Second, we calculate the amplitude and direction of the micromotion of each ion and verify the stability of the ion crystal by time evolving the full equations of motion (i.e., without taking the pseudopotential approximation, similar to Refs. [5, 6])” (line 361).

(Note that our statement that “To the best of our knowledge, this is also the first full-scale simulation of micro-motion for a 1000-ion clock” was in our previous referee response letter but not in the manuscript itself, so there was no need to correct the manuscript for this statement.)

We also thank the reviewer for their suggestion to try different laser beam propagation directions experimentally for laser cooling, and for pointing out the potential impact of this on the broader 3D ion cloud experimental community. Indeed, these are complex many-body systems, and experimental data is critical for advancing theoretical understanding as well.

-
- [1] A. M. Hankin, E. R. Clements, Y. Huang, S. M. Brewer, J.-S. Chen, C. W. Chou, D. B. Hume, and D. R. Leibrandt, Systematic uncertainty due to background-gas collisions in trapped-ion optical clocks, *Phys. Rev. A* **100**, 033419 (2019).
 - [2] N. Huntemann, C. Sanner, B. Lipphardt, C. Tamm, and E. Peik, Single-ion atomic clock with 3×10^{-18} systematic uncertainty, *Phys. Rev. Lett.* **116**, 063001 (2016).
 - [3] S. Brewer, J.-S. Chen, A. Hankin, E. Clements, C. Chou, D. Wineland, D. Hume, and D. Leibrandt, $^{27}\text{Al}^+$ quantum-logic clock with a systematic uncertainty below 10^{-18} , *Phys. Rev. Lett.* **123**, 033201 (2019).
 - [4] G. Gabrielse, X. Fei, L. A. Orozco, R. L. Tjoelker, J. Haas, H. Kalinowsky, T. A. Trainor, and W. Kells, Thousandfold improvement in the measured antiproton mass, *Phys. Rev. Lett.* **65**, 1317 (1990).
 - [5] A. Poindron, J. Pedregosa-Gutierrez, C. Jouvét, M. Knoop, and C. Champenois, Non-destructive detection of large molecules without mass limitation, *J. Chem. Phys.* **154**, 184203 (2021).
 - [6] A. Poindron, J. Pedregosa-Gutierrez, and C. Champenois, Thermal bistability in laser-cooled trapped ions, *Phys. Rev. A* **108**, 013109 (2023).

Reviewer #1 (Remarks to the Author):

I have read the answers to the questions, and the authors answered in great detail. Now the manuscript is acceptable. This Sn^{2+} based many-ion optical atomic clock will be of great interest as a new promising optical clock candidate with sub- $E-19$ inaccuracy.

Reviewer #2 (Remarks to the Author):

dear editor,
after this second revision, I think the paper is ready for publication,
best regards